# Impacts of estimated plume rise on PM$_{2.5}$ exceedance prediction during extreme wildfire events: A comparison of three schemes (Briggs, Freitas, and Sofiev)

Yunyao Li[1,2], Daniel Tong[1,2,3], Siqi Ma[1,2], Saulo R. Freitas[4], Ravan Ahmadov[5,6], Mikhail Sofiev[7], Xiaoyang Zhang[8], Shobha Kondragunta[9], Ralph Kahn[10], Youhua Tang[2,3], Barry Baker[3], Patrick Campbell[3], Rick Saylor[3], Georg Grell[11], Fangjun Li[8]

[1]Department of Atmospheric, Oceanic and Earth Sciences, George Mason University, Fairfax, VA 22030, USA
[2]Center for Spatial Information Science and Systems, George Mason University, Fairfax, VA 22030, USA
[3]Air Resources Laboratory, National Oceanic and Atmospheric Administration, College Park, MD 20740, USA
[4]Center for Weather Forecasting and Climate Studies, National Institute for Space Research, São José dos Campos, 12227, Brazil
[5]Cooperative Institute for Research in Environmental Sciences, University of Colorado at Boulder, Boulder, CO 80305, USA
[6]Global Systems Laboratory, National Oceanic and Atmospheric Administration, Boulder, CO 80305, USA
[7]Atmospheric Composition Research, Finnish Meteorological Institute, Helsinki, 00101, Finland
[8]Geospatial Sciences Center of Excellence, Department of Geography & Geospatial Sciences, South Dakota State University, Brookings, 57007, USA
[9]Satellite Meteorology and Climatology Division, National Oceanic and Atmospheric Administration, College Park, MD 20740, USA
10Earth Sciences Division, National Aeronautics and Space Administration Goddard Space Flight Center, Greenbelt, 20771, USA
[11]Global Systems Laboratory, National Oceanic and Atmospheric Administration, Boulder, CO 80305, USA

*Correspondence to*: Yunyao Li (yli74@gmu.edu), Daniel Tong (qtong@gmu.edu)

**Abstract.** Plume height plays a vital role in wildfire smoke dispersion and the subsequent effects on air quality and human health. In this study, we assess the impact of different plume rise schemes on predicting the dispersion of wildfire air pollution, and the exceedances of the National Ambient Air Quality Standards (NAAQS) for fine particulate matter (PM$_{2.5}$) during the 2020 western United States Wildfire season. Three widely used plume rise schemes (Briggs 1969, Freitas 2007, Sofiev 2012) are compared within the Community Multiscale Air Quality (CMAQ) modelling framework. The plume heights simulated by these schemes are comparable to the aerosol height observed by the Multi-angle Imaging SpectroRadiometer (MISR) and Cloud-Aerosol Lidar and Infrared Pathfinder Satellite Observations (CALIPSO). The performance of the simulations with these schemes varies by fire case and weather conditions. On average, simulations with higher plume injection heights predict lower AOD and surface PM$_{2.5}$ concentrations near the source region but higher AOD and PM$_{2.5}$ in downwind regions due to the faster spread of the smoke plume once ejected. The two-month mean AOD difference caused by different plume rise schemes is approximately 20-30% near the source regions and 5-10% in the downwind regions. Thick smoke blocks sunlight and suppresses photochemical reactions in areas with high AOD. The surface PM$_{2.5}$ difference reaches 70% on the west coast and the difference is lower than 15% in the downwind regions. Moreover, the plume injection height affects pollution exceedance (>35 µg/m$^3$) predictions. Higher plume heights generally produce larger downwind PM$_{2.5}$ exceedance areas. The

PM$_{2.5}$ exceedance areas predicted by the three schemes largely overlap, suggesting that all schemes perform similarly during large wildfire events when the predicted concentrations are well above the exceedance threshold. At the edges of the smoke plumes, however, there are noticeable differences in the PM$_{2.5}$ concentration and predicted PM$_{2.5}$ exceedance region. For the whole studying period, the difference in the total number of exceedance days could be as large as 20 days in northern California, and 4 days in the downwind regions. This disagreement among the PM$_{2.5}$ exceedance forecasts may affect key decision-making regarding early warning of extreme air pollution episodes at local levels during large wildfire events.

40

## 1 Introduction

Wildfires release large amounts of aerosol and trace gases into the atmosphere, which degrades the air quality and adversely affects human health (Koning et al., 1985). Previous studies (Reid et al., 2016; Cascio, 2018) have demonstrated that a strong association exists between exposure to wildfire smoke and all-cause mortality and respiratory morbidity. The global average mortality attributable to landscape fire smoke exposure was estimated at 339,000 deaths annually (Johnston et al., 2012). O'Neil et al. (2021) discuss the regional health impacts of the 2017 Northern California wildfires and estimated 83 excess deaths from exposure to $PM_{2.5}$ (i.e., particles having aerodynamic diameter less than 2.5 μm), of which 47% were attributable to wildfire smoke during the smoke episode. Liu et al. (2021) assessed the health impact of the 2020 Washington State wildfire smoke episode, which caused 38.4 more all-causes mortality cases and 15.1 more respiratory mortality cases. Aerosols emitted from wildfires also affect photolysis rates and photochemistry (Tang et al., 2003) as well as ozone photochemical production (Val Martín et al., 2006; Akagi et al., 2013). Wilmot et al. (2022) produced a decadal-scale wildfire plume rise climatology for the U.S. west coast and Canada and found trends toward enhanced plume heights, and the surface smoke injection to the free troposphere, which suggest a growing impact of wildfires on air quality and regional climate.

Previous studies have found that the smoke injection height plays a vital role in smoke dispersion, as wind speed and direction generally vary with altitude (e.g., Mallia et al., 2018; Vernon et al., 2018;). In addition, a higher injection height will reduce near-source concentration, increase downwind concentrations (Li et al., 2020), and can influence the removal processes and atmospheric lifetime of emitted particles and trace gases. Briggs (1969) introduced a set of semi-empirical formulas to estimate plume injection height for stack emissions from stationary power-plant point sources in different atmospheric stability states using buoyancy flux, horizontal wind speed, static stability, and atmospheric turbulence conditions. This scheme is widely used in dispersion models such as the National Oceanic and Atmospheric Administration (NOAA) Hybrid Single-Particle Lagrangian Integrated Trajectory model (HYSPLIT; Draxler and Hess, 1998) and Community Multiscale Air Quality Modeling System (CMAQ; Byun and Schere, 2006). However, the Briggs scheme was not designed for irregular occurrence large point source emissions, such as forest fires. Also, some of the input parameters, such as heat flux, are difficult to obtain. Freitas et al. (2007) developed a 1-D plume rise and transport parameterization for low-resolution atmospheric chemistry models, which was built upon governing equations for the first law of thermodynamics, vertical motion, and continuity for the water phases. Sofiev et al. (2012) developed a new plume rise scheme, which utilizes fire radiative power (FRP), planetary boundary layer (PBL) height, and the Brunt-Vaisala frequency in the free troposphere to estimate the plume injection height from wildfires. The parameters of the new scheme were determined using the plume height observations collected by the Multi-angle Imaging SpectroRadiometer (MISR) Plume Height Project (Kahn et al., 2008; Mazzoni et al., 2007) in North America (Val Martin et al., 2010) and Siberia. The plume height estimation in models is of great uncertainty. Sessions et al. (2011) tested the Freitas plume rise scheme with Weather Research and Forecasting and Chemistry (WRF-Chem) model and found that differences in injection heights produce different transport pathways. Roy et al. (2017) compared the simulated

plume heights from two different approaches, Western Regional Air Partnership's (WRAP) plume model and the Freitas plume model. Results show that the Freitas plume model got a better diurnal variation of the plume rise height. Mallia et al. (2018) tested different ways to distribute the fire emissions vertically for prescribed fires. Results indicated that plume height plays a critical role in determining how smoke distributes downwind of the fire. Ye et al. (2021) compared the calculated plume heights from 12 state-of-the-art air quality forecasting systems during the Williams Flats fire in Washington State, US, in August 2019, during the Fire Influence on Regional to Global Environments and Air Quality (FIREX-AQ) field campaign. They found that there was a large spread of the modelled plume heights.

In the summer and early autumn of 2020, the western United States (U.S.) experienced a record-breaking wildfire season. A series of large wildfires fuelled by accumulated biomass, heatwaves, and dry winds, burned more than 10 million acres (National Interagency Fire Center, 2020). From late August to early October 2020, the West Coast wildfires contributed 23% of surface $PM_{2.5}$ pollution nationwide and caused 3,720 observed $PM_{2.5}$ exceedances (daily $PM_{2.5}>35\mu g/m^3$ based on National Ambient Air Quality Standards; Li et al., 2021). The thick fire smoke that originated in California, Oregon, and Washington was injected into the free troposphere and transported across the country by the prevailing wind, which caused hazy days (indicated by the high AOD region) in 19 states (Figure 1).

This study aims to evaluate the impact of different plume rise schemes on aerosol distribution and photochemistry during the 2020 record-breaking wildfire season. We use the George Mason University (GMU) wildfire forecast system (Li et al., 2021) that relies on satellite estimates of biomass burning emissions and the CMAQ to simulate the emission, transport, and transformation of smoke during the 2020 summer wildfire season. Three plume rise schemes are used: Briggs (1969), Freitas (2007), and Sofiev (2012). The Briggs (1969) scheme was implemented in the standard release of the CMAQ version. Li et al. (2021) implemented the Sofiev scheme into CMAQ, and in this work, the Freitas scheme is also implemented. The plume injection height's impact on $PM_{2.5}$ vertical distribution is evaluated in section 3.2. Its impact on aerosol optical depth (AOD) and photochemistry is discussed in section 3.3. Finally, we discussed plume rise impact on surface pollution level and $PM_{2.5}$ exceedance in section 3.4.

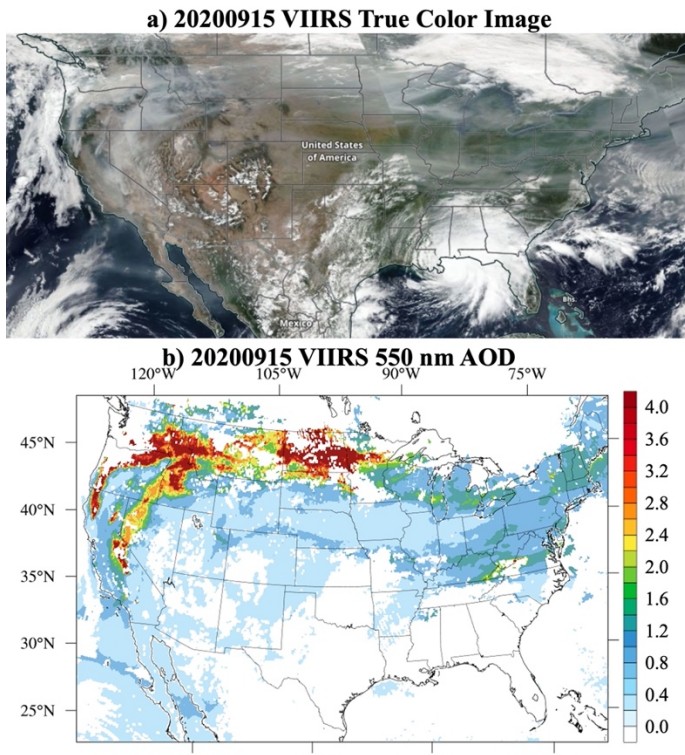

**Figure 1. Observations of wildfire smoke on September 15, 2020, over the continental United States by the Visible Infrared Imaging Radiometer Suite (VIIRS) aboard the Suomi-NPP satellite: a) true color image and b) 550 nm aerosol optical depth (AOD).**

## 2 Methods

### 2.1 Experiment Design

Biomass burning is an important source of global aerosols that have a great impact on air quality. Figure 2 shows how wildfire smoke affects local and downwind air quality (Koppmann et al., 2005; Seinfeld and Pandis, 2016; Schlosser et al., 2017). Wildfire emissions include primary aerosols (direct emission) and large amounts of gases that can be oxidized to form secondary aerosols (generated after emission). In the biomass burning input of our model, the major components of the primary aerosols are organic carbon, non-carbon organic matter, elemental carbon, chloride, and potassium. The other wildfire emissions like $SO_2$, $NO_x$ ($NO+NO_2$), $NH_3$, and VOCs may form secondary aerosols such as sulfate, nitrate, ammonium, and secondary organic aerosols (SOA) after being emitted. The temporal and spatial impacts of plume rise on different primary or secondary aerosol species may be different, as the generation of the secondary aerosols usually takes time. The difference in the dispersion of primary and secondary aerosols will contribute to further differences in photochemistry and health impacts. Therefore, it is important to discuss the impact of plume rise on each primary and secondary aerosol species.

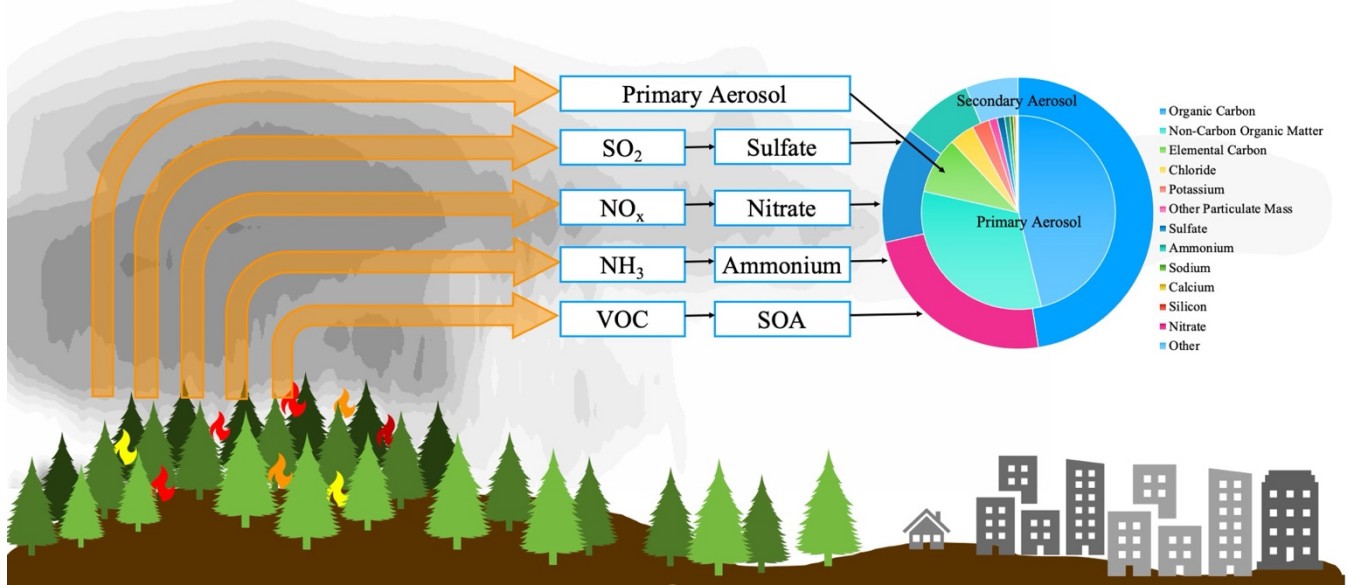

Figure 2[*,+]: **Wildfire primary emissions and downwind evolution.**

[*]Note: The percentage for primary aerosols is from the CMAQ biomass burning input file. The percentage for the secondary aerosols is not real, just for illustration purposes.

[+]Note: The CMAQ model separates organic carbon (OC) and non-carbon elements (O, H, etc) in organic matter (OM).

To evaluate the impact of different plume rise schemes on aerosol dispersion and photochemistry modelling during the 2020 recording-breaking wildfire season, four CMAQ simulations were conducted. In the first run (B69), we used the CMAQ default plume rise scheme based on Briggs (1969). In the second run (F07), we implemented the Freitas et al. (2007) scheme into the CMAQ model and used it to calculate the plume injection height. In the third run (S12), we used the Sofiev et al. (2012) plume rise scheme as implemented in Li et al. (2021). In the fourth run (NoFire), we turned off all types of biomass burning emissions. Wildfire impact is represented by the difference between the simulation with fire and the NoFire run. More information on the three plume rise schemes is provided in section 2.3. Besides the difference in the plume rise scheme, the setups for these three runs are the same. More details about the CMAQ setup are given in section 2.2. Comparing results from these three simulations elucidates the impacts of plume injection height predictions on the distribution of each aerosol species (section 3.2), AOD, and photochemistry (section 3.3), as well as surface air quality and PM$_{2.5}$ exceedances (section 3.4).

**2.2 Description of the Model System**

The George Mason University (GMU) air quality modeling system was employed to simulate the 2020 summer wildfire season from August 1$^{st}$ to September 30$^{th}$, 2020 over the contiguous United States (CONUS) domain. This system uses CMAQ Version 5.3.1 (U.S. EPA, 2020a) as the chemical transport model and the Weather Research and Forecasting (WRF;

Skamarock et al., 2019) model V4.2 output as the meteorology inputs for the CMAQ model. The model domain is configured with 12 km by 12 km horizontal resolution and 35 vertical layers (the same horizontal and vertical resolution as NOAA's operational National Air Quality Forecasting Capability). The initial and boundary conditions for WRF are from the Global Data Assimilation System (GDAS) 0.25-degree analysis and forecast. The main physics choices were the Grell-Freitas scheme (Grell & Freitas, 2014) for parameterized cumulus processes, the Mellor-Yamada-Janjic scheme (Janjic, 1994) for planetary boundary layer (PBL) processes, the two-moment Morrison microphysics (Morrison et al., 2009) for cloud physics processes, the RRTMG scheme (Iacono et al., 2008) for longwave and shortwave radiation, and the Noah scheme (Koren et al., 1999) for land surface processes. The biomass burning emissions product used in this study is the 0.1 degree daily blended Global Biomass Burning Emissions Product from Moderate Resolution Imaging Spectroradiometer (MODIS) and Visible Infrared Imaging Radiometer Suite (VIIRS) (GBBEPx V3, Zhang, et al., 2012, 2019). The GBBEPx Fire radiative power (FRP) in the GBBEPx is averaged from observations from MODIS on Terra and Aqua MODIS and VIIRS M-band on Suomi National Polar-orbiting Partnership (SNPP) and the Joint Polar Satellite System-1 (JPSS) VIIRS. A climatological diurnal cycle from the Western Regional Air Partnership (WRAP) work was applied to the daily GBBEPx emission to derive hourly model-ready emission input. Anthropogenic emissions were prepared with the 2016v1 Emissions Modelling Platform, using the baseline emissions taken from the National Emissions Inventory (NEI) 2016 Collaborative (Eyth et al., 2020). We then shifted the base year emission to the prediction year 2020 using representative days of each month (U.S. EPA, 2020b). The model-ready emission files are processed and generated by the Sparse Matrix Operator Kernel Emissions (SMOKE) model (Houyoux et al., 2000) V4.7. The CB6 gas-phase chemical mechanism (Luecken et al., 2019), AE07 aerosol scheme (Pye et al., 2015; Xu et al., 2018), and aqueous chemistry (Fahey et al., 2017) is used in the CMAQ system. Details about the system setup are shown in Table S1 of Li et al. (2021).

The evaluation of the model performance with Sofiev et al. (2012) plume rise scheme has been discussed by Li et al. (2021). The average correlation between observed (from AirNow) and simulated daily $PM_{2.5}$ concentrations is 0.55. The averaged normalized mean error of the simulated surface $PM_{2.5}$ is 3.9% for the year 2020. The average area hit ratio for exceedances is 0.68 (Fig 2a from Li et al., 2021). A high area hit ratio represents a good capture of the region impacted by smoke. During the peak pollution days (from September $12^{th}$ to $16^{th}$), the area hit ratios were higher than 0.96 with a maximum of 1.0 on September 13th, 2020. This suggests that the model could predict more than 96% of the observed exceedances when the smoke pollution was at its peak. Also, the simulated $PM_{2.5}$ vertical profiles in the West Coast and Central U.S. matched the vertical profiles of backscatter from Cloud-Aerosol Lidar and Infrared Pathfinder Satellite Observations (CALIPSO) near the wildfire source region and downwind area (Fig S2 from Li et al., 2021). Overall, the model can reproduce wildfire smoke dispersion, especially when the smoke is thick.

## 2.3 Description of the plume rise schemes

Three plume rise schemes are used in this study: Briggs (1969), Freitas (2007), and Sofiev (2012).

### 2.3.1 Briggs scheme (B69)

The default plume rise scheme in CMAQ is based on Briggs (1969) and has been modified by revisions in Briggs (1971, 1972, 1984). It uses a set of semi-empirical formulas to estimate plume injection height ($H_p$) in different atmospheric stability states (i.e., neutral, stable, and unstable). Heat flux (B), horizontal wind speed (U), static stability (S), and friction velocity ($x^*$) are used to estimate the plume injection height:

$$H_p = \begin{cases} 1.33 \times BU^{-1}x^{*-2}, neutral \\ 2.6 \times (BU^{-1}S^{-1})^{\frac{1}{3}}, stable \\ 30 \times (BU^{-1})^{\frac{3}{5}}, unstable \end{cases} \tag{1}$$

Previous studies found that FRP is about 10-20% of the total fire heat (Wooster et al., 2005; Freeborn et al., 2008). In this study, we derive the heat flux from FRP provided by the GBBEPx dataset multiplied by a factor of 10 following Val Martin et al. (2012). The Briggs scheme is widely used in chemical transport models; however, it was not designed for forest fires.

### 2.3.2 Freitas scheme (F06)

Freitas et al. (2007) developed a 1-D plume rise and transport parameterization for low-resolution atmospheric chemistry models, which was built upon governing equations for the first law of thermodynamics, vertical motion, and continuity for the water phases (Eq. 1-5 from Freitas et al., 2007). It takes in fire information, including fire area and heat flux, as well as atmospheric profile information, including temperature, moisture, density, and wind velocity. The plume top height is defined as the altitude at which the plume is neutrally buoyant, and is approximated as a vertical velocity < 1 m/s. The Freitas scheme is the default plume rise scheme in WRF-Chem and has been widely used in many studies (e.g., Sessions et al., 2011; Val Martin et al., 2012; Roy et al., 2017; Mallia et al., 2018). However, the Freitas scheme has never before been used with CMAQ. In this work, the FRP-based Freitas scheme from High-Resolution Rapid Refresh Smoke (HRRR-Smoke; Ahmadov et al., 2017) model has been implemented into CMAQ. Wind, temperature, pressure, and humidity from WRFV4.2 meteorology inputs as well as FRP and fire burning area are used to calculate the plume injection height in the model. The FRP from GBBEPx and fire size from RAP-Chem (Rapid Fresh-Chemistry; Archer-Nicholls, et al., 2015) are used to calculate fire buoyancy in the model.

### 2.3.3 Sofiev scheme (S12)

Sofiev et al. (2012) developed a new plume rise scheme that considers wildfire plumes in a way similar to Convective Available Potential Energy (CAPE) computations. It utilizes FRP, PBL height ($H_{PBL}$), and the Brunt-Vaisala frequency in the free troposphere ($BV_{FT}$) to estimate the plume injection height from wildfires:

$$H_p = \alpha H_{PBL} + \beta \left(\frac{FRP}{FRP_0}\right)^\gamma \exp\left(-\frac{\delta BV_{FT}^2}{BV_0^2}\right) \tag{2}$$

Where $FRP_0$ is the reference fire radiative power which equals $10^6$ W, $BV_0$ is the reference Brunt-Vaisala frequency which equals $2.5 \times 10^{-4}$ s$^{-2}$, and $\alpha$, $\beta$, $\gamma$, $\delta$ are constants. In our previous study, we added the Sofiev scheme to CMAQ (Li et al., 2021)

and applied it previously to predict air quality during the 2020 wildfire season with the same constants from Sofiev at al. (2012).

After getting the estimated plume injection height from the three schemes, the fire emissions were distributed between 0.5-1.5 times plume injection height (default setting in CMAQ). The three schemes used in the current experiment are very different in their nature and underlying assumptions, but they all were developed with an individual fire as a model source of buoyancy and smoke. In this experiment, as well as in many other applications of these schemes, the input fire information is gridded with a grid cell size of several km or larger. Strictly speaking, such a setup goes beyond the area of applicability of these schemes. However, with a growing number of gridded fire emission products and their applications for atmospheric composition and air quality tasks, it is important to evaluate this very setup – and to compare the robustness of these models to the violation of their underlying assumptions. In this study, we use the default coefficient settings in each scheme. We did not tune the coefficient of any scheme to get the best simulation for any major fire case. The main focus of this study is to evaluate the impact of different plume injection heights on the near source and downwind air quality, and the two-month average state is more important to our results and future health studies.

## 2.4 Observation data

### 2.4.1 MISR and CALIPSO plume height observation

The predicted plume height is evaluated using observations from Multi-angle Imaging SpectroRadiometer (MISR) and Cloud-Aerosol Lidar and Infrared Pathfinder Satellite Observations (CALIPSO). The MISR instrument obtains imagery of each location within its 380 km-wide swath at nine view angles, ranging from 70° forward, through the nadir, to 70° aft, along the orbit track, in each of four spectral bands centered at 446 (blue), 558 (green), 672 (red), and 866 nm (near-infrared) wavelengths (Diner et al., 1998). MISR is in a sun-synchronous orbit, crossing the Equator at approximately 10:30 AM local time, so observations in the study region occurred in the mid-to-late morning, The MISR INteractive eXplorer (MINX) software is used in this study to derive plume heights from MISR imagery (Nelson et al., 2013; Val Martin et al., 2018). The MINX wind-corrected plume height information is then used to evaluate the simulated plume height in this paper.

CALIPSO is an Earth Science observation mission that was launched on 28 April 2006 and flies in a nominal orbital altitude of 705 km and an inclination of 98 degrees as part of a constellation of Earth-observing satellites. CALIPSO's lidar instrument, the Cloud-Aerosol Lidar with Orthogonal Polarization (CALIOP), provides high-resolution vertical profiles of aerosol and cloud attenuated backscatter signals at 532 nm and 1064 nm (Winker et al., 2007). The footprint of the lidar beam has a 100 m cross-section with an overpass around 1:30 p.m. local time. The CALIPSO smoke injection heights are directly calculated from Level 1 attenuated backscatter profiles at 532 nm following Amiridis et al. (2010). There are several steps involved in this process. First, GBBEPx FRP data were used to locate the fire location along the CALIPSO swath. Then, a slope method

(Pal et al., 1992) is applied to each profile to smooth out the original Level 1 532 nm attenuated backscatter coefficient profiles at each fire point. Next, we calculate the steep gradient in the attenuated backscatter profiles. The height of the minimum gradient value is selected as the smoke injection height.

### 2.4.2 AirNow surface PM$_{2.5}$ data

The hourly ground PM$_{2.5}$ observations from the U.S. EPA AirNow network are used to evaluate the surface air pollution predictions in this study. The real-time AirNow measurements are collected by the state, local, or tribal environmental agencies using federal references or equivalent monitoring methods approved by EPA. It contains air quality data for more than 500 cities across the U.S., as well as for Canada and Mexico.

### 2.4.3 VIIRS AOD data

The simulated AOD results are compared to the VIIRS Enterprise AOD from Suomi-NPP (Zhang et al., 2016; Kondragunta et al., 2017). The VIIRS Enterprise Aerosol Algorithm retrieves AOD at the 750-m pixel level for the nominal wavelength of 550 nm using radiances from 11 VIIRS channels (412, 445, 488, 555, 672, 746, 865, 1240, 1378, 1610, and 2250 nm). The AOD is calculated separately for land and ocean using a lookup table of pre-computed values for several atmospheric parameters to simplify radiative transfer calculations.

## 3 Results

### 3.1 Comparing simulated plume heights against MISR and CALIPSO observations

The simulated plume heights from three simulations: B69, F07, and S12 are compared with MISR observations for the Milepost 21 Fire on August 15[th], 2020, and the August Complex Fire on August 31[st], 2020 (Figure 3) at the MISR local overpass time of around 19 UTC. The smoke heights from model 3-dimensional fields were interpolated to the MISR observation pixels using the nearest neighbour approach. The performance of different schemes varies by fire cases and weather conditions. For the Milepost 21 Fire, the plume heights simulated by B69 and S12 are similar but 25% and 3% lower than that by F07 for the easterly and westerly plume. In the case of the August Complex Fire northerly plume, the plume heights simulated by S12 are 4% and 8% higher than that by B69 and F07, respectively. For the southerly plume, the plume heights simulated by B69 are 16% and 5% higher than that by F07 and S12. The simulated PBL heights were displayed in Figure 3 as a reference. When the fire injection height is lower than the PBL height, the pollution could become confined in the PBL (Sofiev et al. 2021; Thapa et al, 2002). However, when the plume height is higher than PBL, the fire smoke can be dispersed into the free troposphere where wind speeds are stronger, leading to a wider range of pollution dispersion. In all four cases analyzed in Figure 3, the simulated plume heights from the three schemes surpassed the model PBL. Previous studies found that for large fires that are injected above the PBL, the plume height calculated by S12 is less sensitive to FRP than by B69 (Li et al., 2020). Some of the fire points during the August Complex Fire had higher FRP than that during the Milepost 21 Fire, so the estimated plume

height by B69 is higher than that by S12. For the F07 scheme, the plume injection height is higher when it is wetter (Freitas et al., 2007). The water vapor mixing ratio on August 15th is higher than on August 31st, which contributes to the higher plume height during the Milepost 21 Fire than during the August Complex fire. According to the box and whisker chart shown in the right panel of Figure 3, the simulated plume heights are all within the range of MISR observation. Overall, the simulated plume heights with all three schemes are reasonably comparable to the MISR observations.

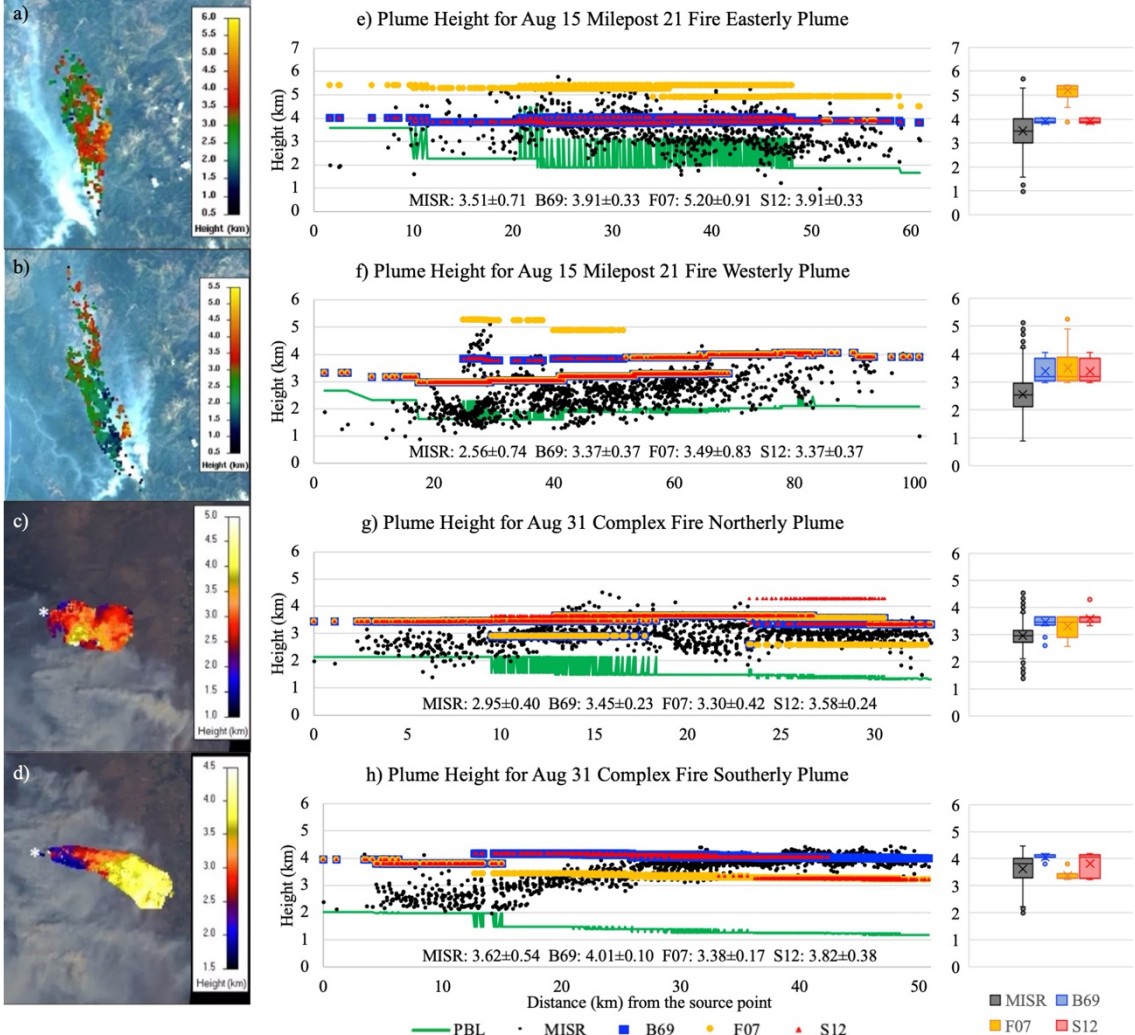

**Figure 3: MISR plume heights superposed on the MODIS Terra visible images** (a-d) and the comparisons of the observed plume height with the simulated plume heights (e-h) for the August 15th Milepost 21 fire easterly plume (a, e), westerly plume (b, f), August 31st Complex fire northerly plume (c, g), and southerly plume (d, h).**


**Source: MISR Active Aerosol Plume-Height (AAP) Project / R.A. Kahn, K.J. Noyes, J. Limbacher (NASA Goddard Space Flight Center), A. Nastan (JPL-Caltech), J. Tackett, J-P. Vernier (NASA Langley Research Center)**

The vertical profiles of CMAQ simulated PM2.5 are also compared to the CALIPSO daytime aerosol vertical profile. The daytime CALIPSO overpass occurs around 1:30 p.m. local time, closer to the peak fire behaviour in the afternoon than the MISR observations. Figure 4 shows the comparison between the CALIPSO plume height results and the estimated plume heights from the three plume rise schemes for west coast fires. The mean bias for the three schemes is -0.60 for B69, -0.67 for

F07, and 0.13 for S12. In most cases, the plume heights from the three schemes are close to each other, especially for the cases with plume tops under 4 km. For strong fires with plume tops higher than 4 km, S12 seems to be more skillful than B69 and F07. It successfully simulates the high plume top observed by CALIPSO, whereas B69 and F07 tend to underestimate the plume heights.

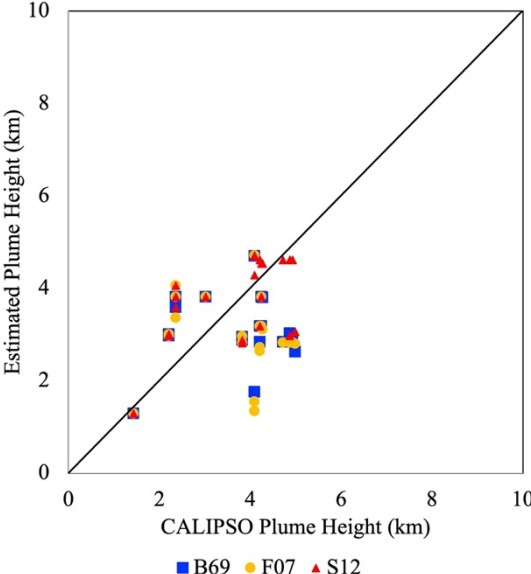

**Figure 4: Comparisons of plume top heights from three simulations: B69 (blue rectangle), F07 (orange dot), and S12 (red triangle), against aerosol height observations from the CALIPSO for west coast fires.**

### 3.2 Impact of estimated plume rise on PM2.5 vertical distribution

In this section, we investigate the impact of plume injection height on different PM2.5 chemical components. Figure 5 shows the vertical profile of the two-month average PM2.5 concentration for the three experiments. Over the two months, B69

simulated a higher average plume height and injected more PM2.5 in the free troposphere than F07 and S12. Meanwhile, F07 simulated a lower average plume height and therefore keep more PM2.5 in the boundary layer than B69 and S12.

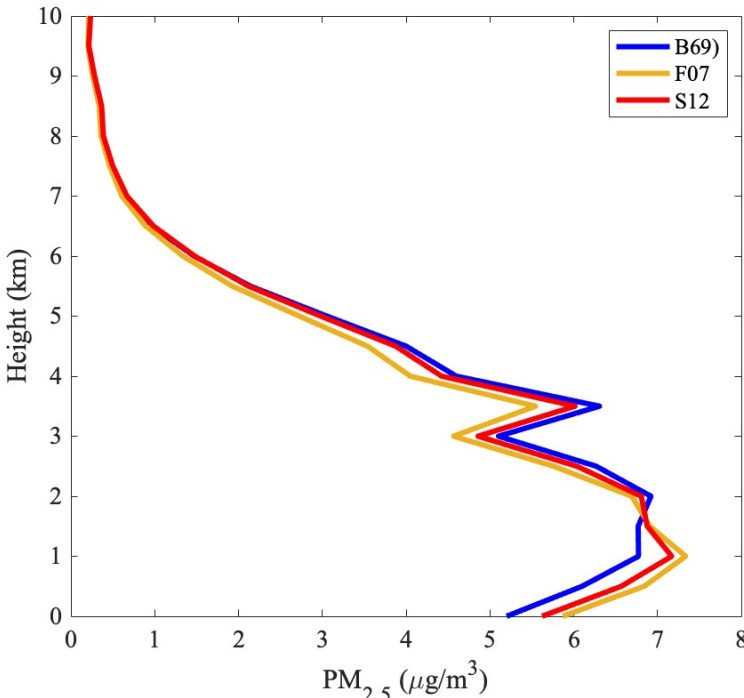

**Figure 5: Vertical profile of two-month average PM2.5 concentration for B69, F07, and S12 in the CONUS domain.**

Figure 6 shows the distribution of simulated $PM_{2.5}$ components from both direct emissions and secondary formation (the impact of other $PM_{2.5}$ sources was removed by subtracting the results of NoFire run) from B69, F07, and S12 in three different regions: the western U.S. (west of 120° W), the central U.S. (between 120° W and 100° W), and the eastern U.S. (between 100° W and 80° W). For all three schemes, organic matter (OM) dominates the chemical composition of $PM_{2.5}$, at 63%-64% near the source region in the western U.S., remains dominant in the downwind regions at ~ 61% between 120° W and 100° W, and 57%

between 100° W and 80° W. A high OM portion highlights the predominant effect of wildfire emissions on air quality during the Gigafire period (Li et al., 2021). The second most abundant component is nitrate ($NO_3$) at 11% - 12% near the source region and 13% - 16% in the downwind region. A higher portion of $NO_3$ in the downwind region than in the source region reflects the decreased contribution of primary aerosols, and increases in secondary aerosols. The other component with a similar spatial gradient is ammonium ($NH_4$), which contributes 3% to $PM_{2.5}$ near the source region and 5% - 6% in the

downwind region. Elemental carbon contributes 10% to $PM_{2.5}$ concentration near the source region and 8% - 9% in the downwind region. Potassium (K), a fingerprint element to indicate fire contribution, accounts for 3% of $PM_{2.5}$ near the source region and 2% ~ 3% in the downwind region. Sulfate ($SO_4$) contributed 3% near the source region and 6% - 8% in the downwind region. In summary, $PM_{2.5}$ species that are not significantly affected by secondary aerosol formation, such as elemental carbon and potassium, have their contributions decrease when transported downwind. For the $PM_{2.5}$ species that are

affected by secondary aerosol formation (e.g., nitrate, sulfate, and ammonium), the contribution increases when transported downwind. These results show that the PM2.5 composition, integrated over all vertical layers, is not sensitive to the choice of plume rise scheme.

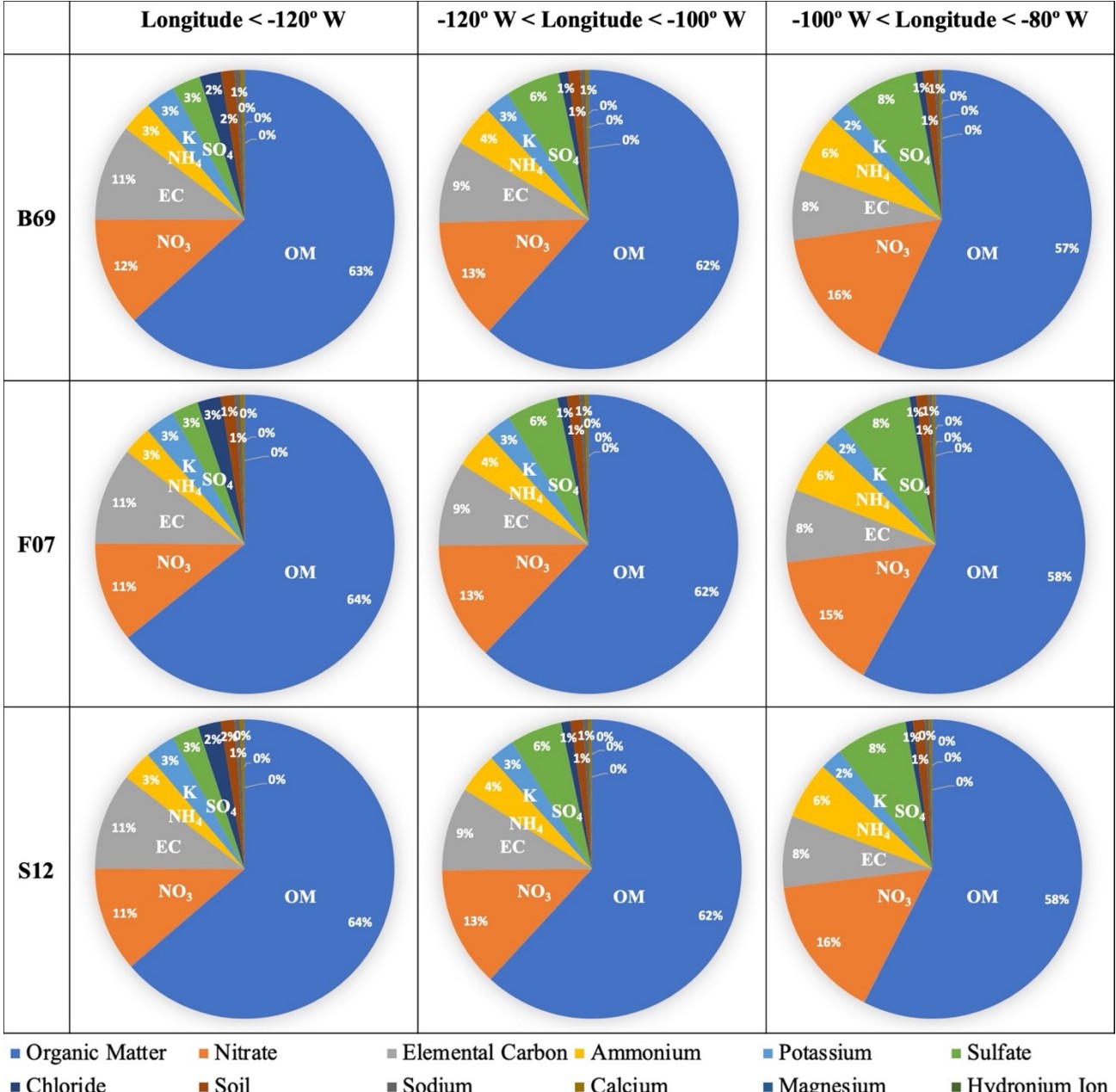

Figure 6: The simulated PM2.5 chemical components (%) with the B69, F07, and S12 plume rise schemes in three
different regions: in the west of 120° W (left), between 120° W and 100° W (middle), and between 100° W and 80° W

**(right). The data are integrated over all vertical layers and averaged during the analysis period. The top 6 components are labeled in each plot.**

Figure 7 shows the difference in the zonal mean (average for each latitude) concentrations of six major $PM_{2.5}$ species (i.e., organic matter, nitrate, elemental carbon, ammonium, potassium, and sulfate) and total $PM_{2.5}$ when using different plume rise schemes over the whole domain. Overall, most of the differences are found over the west coast region (in the west of 115° W). The simulation with B69 produces a higher plume height on average, resulting in greater transport of smoke aloft, and hence higher downwind $PM_{2.5}$ than that with the F07 or S12 schemes. The B69 plume rise scheme has a higher downwind impact and slightly lower near-source impact for $PM_{2.5}$ species that contain secondary aerosols (e.g., organic matter, nitrate, ammonium, and sulfate) than primary aerosols (e.g., elemental carbon and potassium), due to the time required to form secondary aerosols.

Among the three simulations, the largest differences in $PM_{2.5}$ are found from the surface to 8 km over the source region. Over the downwind region, the bulk of $PM_{2.5}$ differences is found in the middle and upper troposphere. In addition, we noticed that the simulations with F07 and S12 produce more $PM_{2.5}$ than that with B69 between 6-8 km during the analysis period. This is because, in the cases of a strong fire, the plume injection height simulated by F07 or S12 could be higher than B69 (e.g., Fig 3e). However, the difference in $PM_{2.5}$ above 6 km is very small compared to those below 6 km. The total $PM_{2.5}$ difference caused by different plume rise schemes is about 30% near the source and 5% in the downwind region. The difference in surface $PM_{2.5}$ have a large impact on surface pollution levels and human health. More discussion on the impact of plume height on surface air quality is presented in section 3.4. Although the upper-level $PM_{2.5}$ difference is expected to have a smaller impact on human health, it may affect cloud formation, photochemical reactions, and the radiative budget in the Earth system.

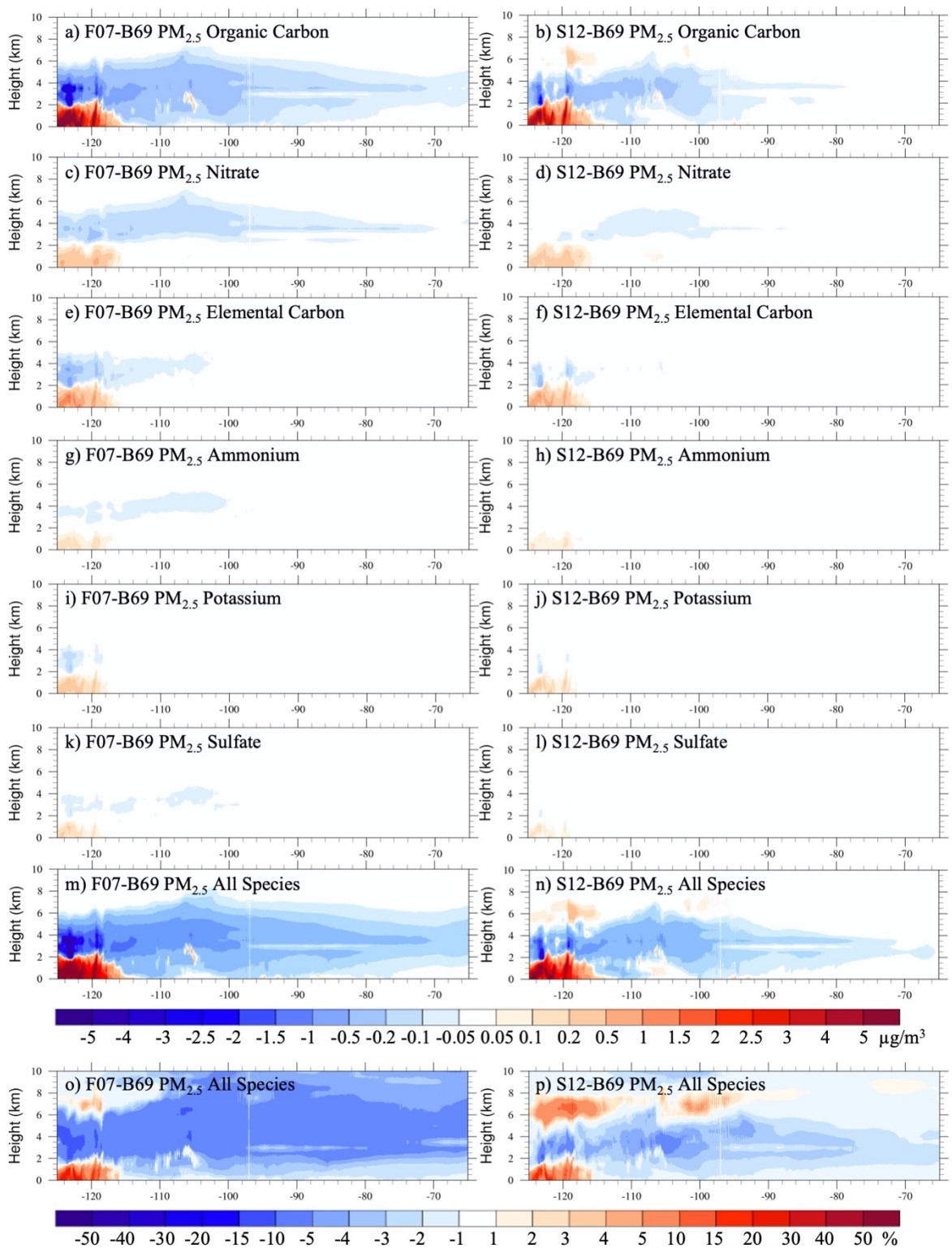

**Figure 7: (a-n) Zonal mean difference in predicted concentrations of six major PM$_{2.5}$ species among the simulations using the B69, F07, and S12 schemes from August 1$^{st}$ to September 30$^{th}$, 2020: Organic Matter (a-b); Nitrate (c-d); Elemental Carbon (e-f); Ammonium (g-h); Potassium (i-j); and Sulfate (k-l). The difference in total PM$_{2.5}$ is displayed by both absolute values (m-n) and percentage (o-p).**

### 3.3 Impact of estimated plume rise on aerosol optical depth and photochemistry

Wildfire smoke increases the aerosol loading in the atmosphere and consequently the AOD over both the source region and downwind regions. According to our previous study (Li et al., 2021), from September 14–17$^{th}$, 2020 smoke from the West Coast was transported to the northeastern part of the U.S. The downwind transport of wildfire smoke is highly dependent on plume rise estimation. Figure 8a shows the two-month averaged AOD from VIIRS compared with model simulations (Figures 7b-d). The CMAQ predicted AOD was interpolated to the VIIRS pixels that passed quality control using the nearest neighbor approach. When comparing the CMAQ AOD to VIIRS AOD (Figures 7b-d), we applied VIIRS AOD saturation level (AOD ≤5) to CMAQ AOD results (any CMAQ AOD values higher than 5 were changed to 5). In the west coast high peak region, all three runs capture the observed AOD high peak near the San Francisco region, but the simulated AOD peak is lower than VIIRS observed. The average AOD from VIIRS observation is higher than 2. However, among the three CMAQ runs, only F07 simulated an average AOD higher than 2. In the downwind region, all three CMAQ runs reproduce the general downwind transport pattern, but the simulated smoke affected region (AOD>0.5) is smaller than the observations.

Figures 7e-h show the AOD differences and the difference ratio (percentage of the difference relative to B69) between the different plume rise scheme simulations. When comparing different model simulations, the AOD saturation level is removed. Near the source region, F07 and S12 simulate more AOD than B69, a pattern that is the opposite of that for plume rise estimation (lower plume height than that with B69). In the downwind region, B69 simulates more AOD than F07 and S12. The difference is approximately 20-30% over the source region and 5-10% over the downwind region. One possible reason that B69 predicts lower AOD near the source region and higher AOD in the downwind region compared to F07 and S12 is that a higher plume height will inject more aerosol into the free troposphere where the wind speed is stronger, accelerating the dispersion of the fire pollution. Therefore, the higher plume height will lead to lower AOD near the source region but higher AOD in the downwind region. The result is consistent with previous studies (Mallia et al., 2018; Vernon et al., 2018; Li et al., 2020).

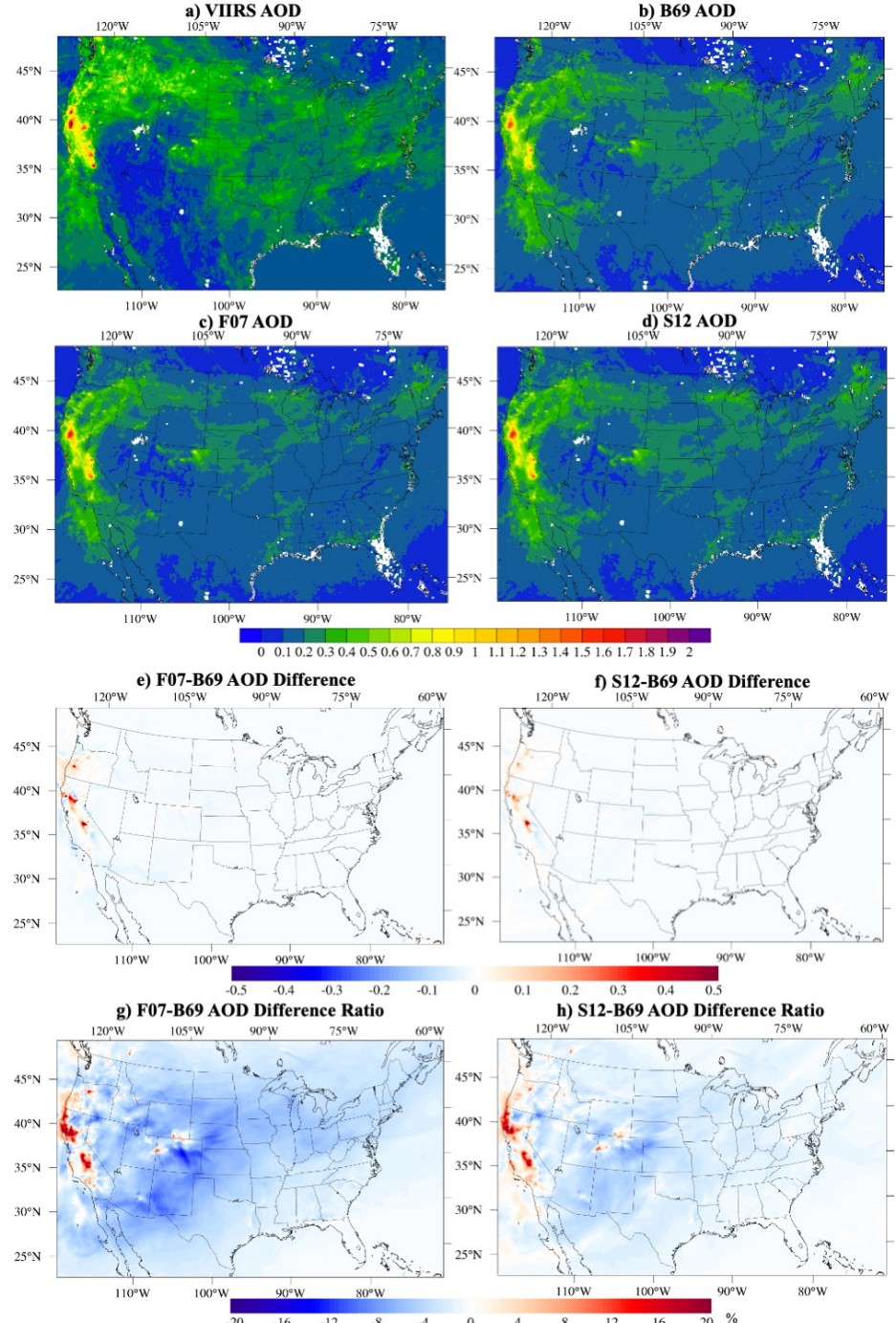

**Figure 8: Two-month average AOD from VIIRS (a), B69 run (b), F07 run (c), and S12 run (d); the average AOD differences between F07 and B69 (e), and between S12 and B69 (f) from August 1ˢᵗ, 2020 to September 30ᵗʰ, 2020; and the average AOD difference ratio between F07 and B69 (g), and between S12 and B69 (h) during the same period.**

The difference in the dispersion of fire pollution caused by the various estimated plume injection heights leads to further differences in the chemistry and photolysis reactions. Previous studies found that the thicker smoke, indicated by higher AOD, may absorb and/or scatter a larger fraction of sunlight, hence affecting photolysis reactions (Dickerson et al., 1997; Castro et al., 2001; Kumar et al., 2014; Baylon et al., 2018). Here, we simply examine how the plume rise differences affects photochemistry by comparing the photolysis rate of $NO_2$ ($NO_2 + h\nu \rightarrow NO + O$) from the three runs, which is a key reaction that leads to the formation of tropospheric ozone. The differences in the $NO_2$ photolysis rate (NO2_IUPAC10, in min$^{-1}$) are shown in Figure 9. Figures 9 a and d show the photolysis rate difference and difference ratio between B69 and the NoFire experiments. The photolysis rate results in the B69 were lower than the NoFire simulation, which proves that fire smoke led to the reduction of the photolysis rate, consistent with the findings of previous studies. The photolysis rate reduction caused by the fire smoke was found in the whole domain, both in the near-source region and the downwind region. Near the fire source, the photolysis rate reduction was more than 50%. Figures b, c, e, and f show the photolysis rate difference and difference ratio between the three experiments with different plume rise schemes. Near the source region where F07 and S12 simulate a higher AOD than B69 (Figure 8), the NO2_IUPAC10 is reduced. Meanwhile, in the downwind region, where F07 and S12 simulate a lower AOD, the photolysis rate is higher than B69. Therefore, the difference in the plume injection height would affect the fire-induced photolysis rate reduction.

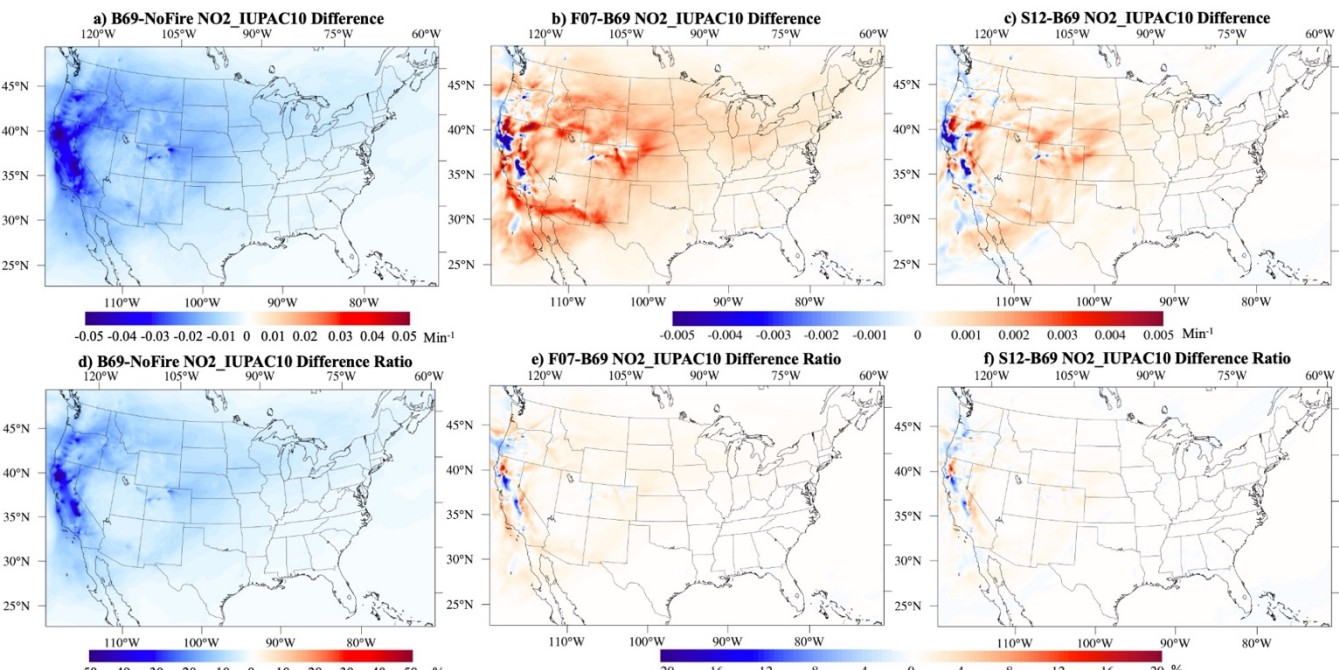

**Figure 9: The average photolysis rate NO2_IUPAC10 differences between B69 and NoFire (a), between F07 and B69 (b), and between S12 and B69 (c) from August 1$^{st}$, 2020 to September 30$^{th}$, 2020; and the average photolysis rate**

 NO2_IUPAC10 difference ratio B69 and NoFire (d), between F07 and B69 (e), and between S12 and B69 (f) during the same period.

### 3.4 Impact of estimated plume rise on surface PM2.5 and exceedance of NAAQS

Surface or ambient $PM_{2.5}$ is the common measure used to link exposure to wildfire smoke to health endpoints such as asthma and chronic obstructive pulmonary disease (Reid et al., 2016). To protect human health and the environment, the National
395  Ambient Air Quality Standards (NAAQS) have been established for $PM_{2.5}$ and other criteria air pollutants ($NO_2$, $O_3$, $SO_2$, CO, $PM_{10}$, and lead). The daily $PM_{2.5}$ NAAQS is 35 $\mu g/m^3$ for the 24-hour mean $PM_{2.5}$ concentration (U.S. EPA, 2020c). The simulated surface $PM_{2.5}$ differences caused by different plume rise schemes are shown in Figure 10. The F07 and S12 simulations, which have averaged lower initial plume heights, yield higher surface $PM_{2.5}$ concentrations than the B69 simulation over the west coast, whereas the opposite patterns are found in the central and the eastern U.S. The surface $PM_{2.5}$
400  difference caused by different plume rise schemes reaches 70% over the west coast, which is much higher than the AOD differences. In the downwind regions, the surface $PM_{2.5}$ difference caused by different plume rise schemes is less than 15%, meaning that the effects of the plume rise estimation on surface $PM_{2.5}$ occur mainly near the source region.

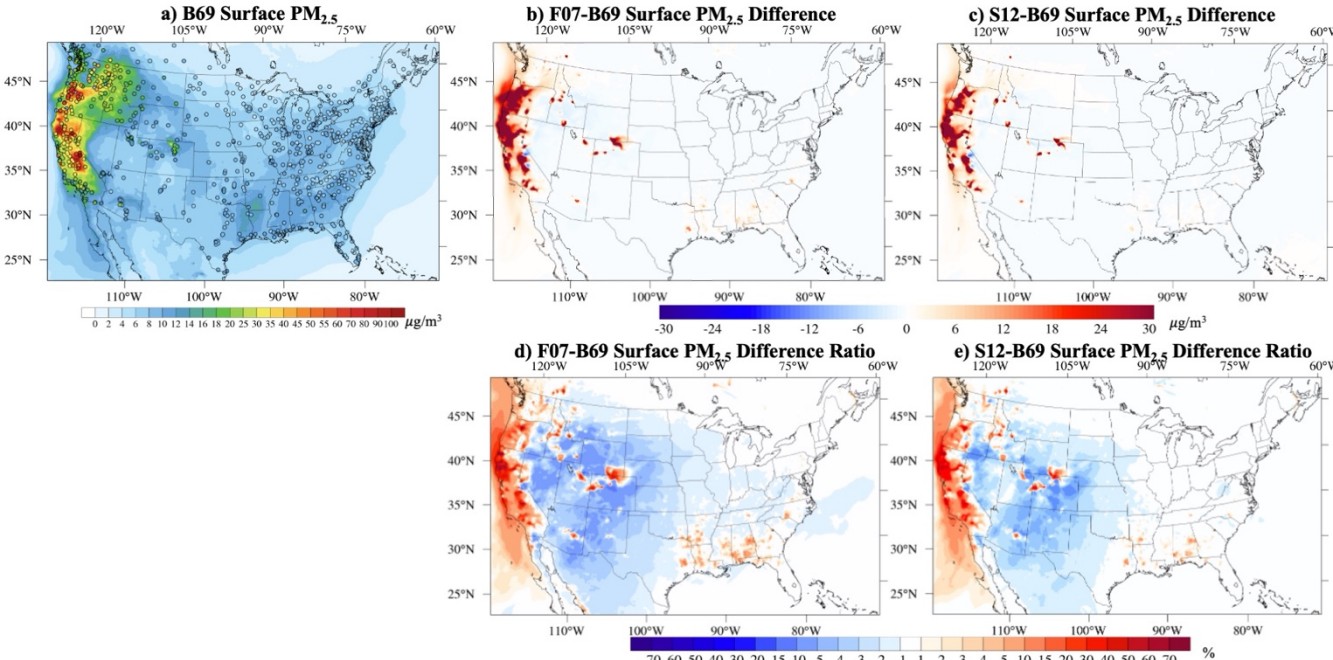

**Figure 10: Simulated and observed surface $PM_{2.5}$ from August 1st, 2020, to September 30th, 2020: a) average surface
405  $PM_{2.5}$ simulated with B69 overlaid by AirNow observations; b) difference in averaged surface $PM_{2.5}$ between F07 and B69; c) difference between S12 and B69; d) and e) same as b) and c), but for the differences in percentage (%) between F07 and B69, and between S12 and B69, respectively.**

Next, we examine how the plume rise estimation affects the prediction of PM$_{2.5}$ exceedances. Figure 11a shows the daily mean surface PM$_{2.5}$ difference between the F07 and B69 runs for Aug 20$^{th}$, 2020 (the first fire peak during the study period). The simulated PM$_{2.5}$ exceedance regions (PM$_{2.5}$>35 $\mu g/m^3$, defined by NAAQS, same level with US EPA defined unhealthy for sensitive groups), unhealthy region (PM$_{2.5}$>55 $\mu g/m^3$, defined by US EPA), and very unhealthy region (PM$_{2.5}$>150 $\mu g/m^3$, defined by US EPA) by different plume rise schemes overlaid by AirNow observed exceedance for the same day are shown in Figures 10b, c, and d. According to Figure 11b and c, on August 20$^{th}$, 2020, B69 and S12 simulated more PM$_{2.5}$ exceedance and a larger unhealthy region in the downwind regions (Wyoming (WY) and Idaho (ID), magenta and blue region), whereas F07 and S12 simulated more exceedance and a larger unhealthy region in the southeastern U.S. (yellow and orange region), where prescribed fires were the major biomass burning sources. In WY and ID, where F07 did not simulate the PM$_{2.5}$ exceedance whereas B69 and S12 did, the difference between F07 and B69 reached 15 $\mu g/m^3$ (Figure 11a). Furthermore, B69 and S12 simulate some very unhealthy regions in Nevada, whereas F07 simulates more very unhealthy regions in central and southern California. Although these schemes agree on the PM$_{2.5}$ exceedance forecast in the majority region, the disagreements in the downwind areas (i.e., ID and WY for this case) may affect key decision-making on early warnings of extreme air pollution episodes at local levels during large wildfire events.

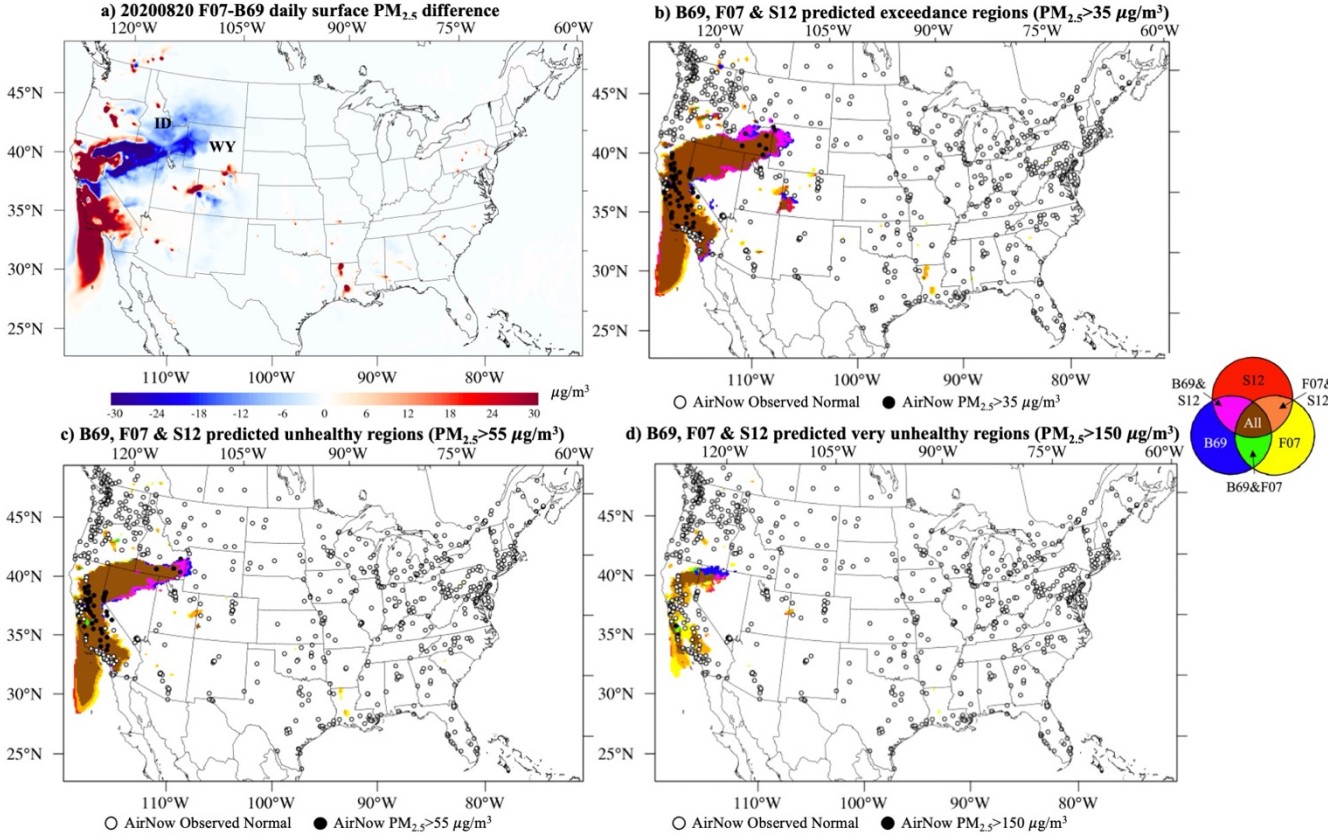

**Figure 11: Predicted surface PM$_{2.5}$ concentrations above unhealthy levels by the S12, F07, and B69 runs for August 20$^{th}$, 2020: a) The daily mean surface PM$_{2.5}$ difference between F07 and B69 runs; b) simulated PM$_{2.5}$ exceedance regions by B69, F07, and S12 overlaid by AirNow observed exceedance (PM$_{2.5}$>35 μg/m$^3$); c) same to b) but for US EPA defined unhealthy regions (PM$_{2.5}$>55 μg/m$^3$); d) same to b) but for US EPA defined very unhealthy region (PM$_{2.5}$>150 μg/m$^3$). The brown color represents the region where the runs with all three schemes simulate PM$_{2.5}$ exceedances; the blue (red/yellow) color represents the region where only B69 (S12/F07) simulates the PM$_{2.5}$ exceedance; the green represents the region where both the B69 and F07 simulate the PM$_{2.5}$ exceedance; the magenta color represents the region where both the B69 and S12 simulate the PM$_{2.5}$ exceedance; the orange represents the region where both F07 and S12 simulate the PM$_{2.5}$ exceedance.**

The total number of predicted exceedance days from the B69 simulation and the differences between B69, F07, and S12 are shown in Figure 12. All the states in the western coast and mountain region experienced at least one day of PM$_{2.5}$ exceedance (Figure 12a). Most of the region in northern California experienced more than 20 exceedance days, with a maximum of more than 35 days. F07 and S12 simulate more exceedance days on the west coast near the source region and in the southeast. The difference in the exceedance days could be as large as 20 days in northern California. B69 simulates more exceedance days in downwind regions such as Nevada, Idaho, Montana, and Wyoming. The difference could reach 4 days in the downwind regions.

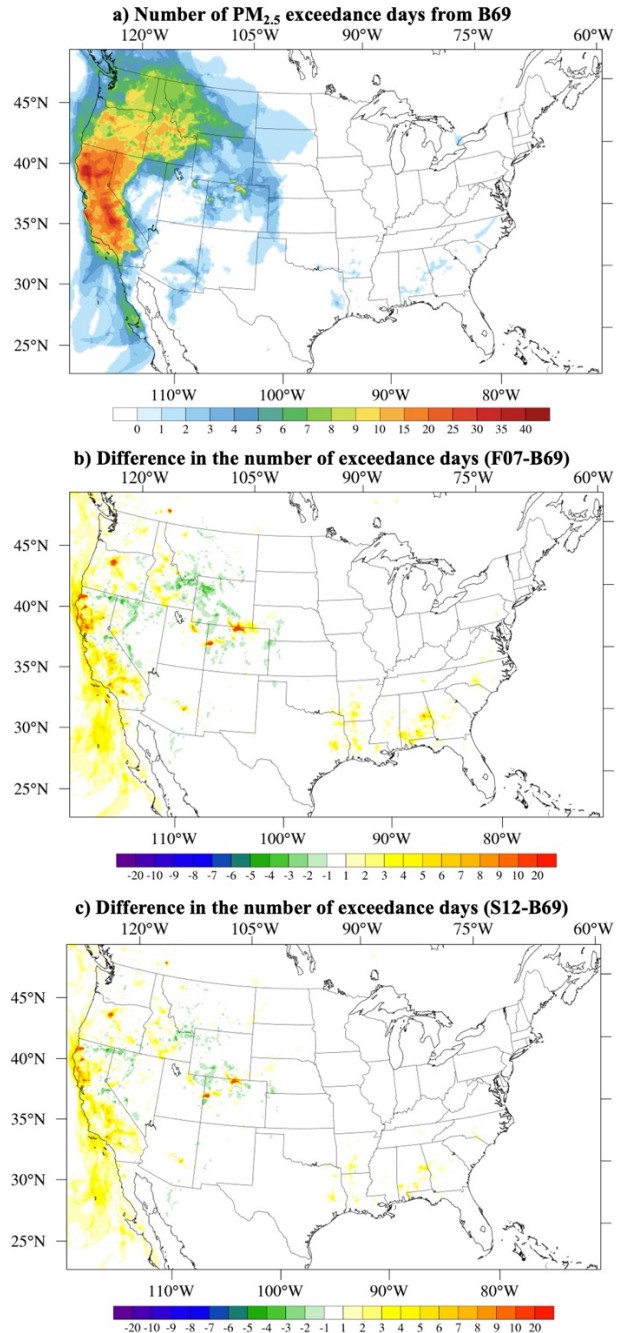


**Figure 12: The CMAQ B69 predicted total number of PM$_{2.5}$ exceedance days during Aug-Sep, 2020 (a); the difference in the number of predicted exceedance days between B69 and F07 (b), and between B69 and S12 (c).**


**4 Conclusion**

In this study, we use CMAQ with three different plume rise schemes, namely Briggs 1969 (B69), Freitas 2007 (F07), and Sofiev 2012 (S12), to understand the impact of plume rise calculation on aerosol and photochemistry during the 2020 western U.S. wildfire season. The plume heights simulated by all three schemes are comparable to MISR and CALIPSO observations

of aerosol height. The performance of the simulations with different schemes varies for different fire cases and weather conditions (i.e., humidity). On average, the B69 predicts higher injection heights than F07 and S12, leading to higher downwind $PM_{2.5}$ concentrations due to the stronger transport at the higher altitude. The largest $PM_{2.5}$ differences are found from the surface to 8 km over the source region. Over the downwind region, the bulk of the $PM_{2.5}$ differences is found in the middle and upper troposphere. The total $PM_{2.5}$ difference is approximately 30% near the source and 5% in the downwind region.

Furthermore, we found that the plume rise scheme has a higher downwind impact and slightly lower near-source impact for $PM_{2.5}$ species that contain secondary aerosols over primary aerosols.

Thick fire smoke also increases AOD in the source and the downwind regions. On average, F07 and S12, which estimate lower plume height, simulate greater smoke AOD near the fire source region than B69. In the downwind region, B69 simulates

higher AOD than F07 and S12. The difference is approximately 20-30% near the source region and 5-10% in the downwind region. When AOD is higher, the thicker smoke may block more sunlight and affect the photolysis reaction rates. Near the source region, where F07 and S12 simulate a higher AOD, the photolysis reaction rate decreases.

Finally, we analyzed the effect of plume rise estimation on the prediction of $PM_{2.5}$ exceedances. The F07 and S12 simulations,

which have lower averaged plume heights, predict higher surface $PM_{2.5}$ concentrations than B69 over the west coast, whereas the opposite patterns are found in the central and the eastern U.S. The effects of the plume rise estimation on surface $PM_{2.5}$ occur mainly near the source region. The surface $PM_{2.5}$ difference caused by different plume rise schemes reaches 70% over the west coast and is less than 15% in the downwind regions. These results suggest that the effects of plume rise estimation on surface $PM_{2.5}$ occur mainly near the source region, whereas in the downwind region, the majority of effects are in the free

troposphere. For $PM_{2.5}$ exceedance prediction, higher plume height produces a larger $PM_{2.5}$ exceedance area in the downwind region. In most affected areas, the predicted $PM_{2.5}$ exceedance regions from the three schemes overlapped. In non-overlapping regions, the simulated differences in $PM_{2.5}$ could reach 15 $\mu g/m^3$. For the whole studying period, the difference in the total number of exceedance days could be as large as 20 days in northern California, and 4 days in the downwind regions. F07 and S12 simulated more exceedance days near the fire source region, while B69 simulates more exceedance days in downwind

regions such as Nevada, Idaho, Montana, and Wyoming. Such $PM_{2.5}$ exceedance forecast differences may affect key decision-making on early warnings of extreme air pollution episodes at local levels during large wildfire events.

The WRF-CMAQ system used in this study is an offline model. The heat emitted by the fire calculated in the CMAQ does not influence the meteorology model (WRF), such as the PBL height, temperature and wind field. In the future, online models will be utilized to further study the plume rise estimation impacts on air quality.

*Code availability*. CMAQ documentation and released versions of the source code are available on the U.S. EPA modeling site https://www.cmascenter.org/. The source code of WRF is available at https://github.com/wrf-model/WRF.

*Data availability*. The MISR data used in this paper can be found here: http://air.csiss.gmu.edu/yli/paper_data/MISR. The GBBEPx data can be downloaded from https://satepsanone.nesdis.noaa.gov/pub/FIRE/GBBEPx-V3/. The VIIRS measurements can be downloaded from http://air.csiss.gmu.edu/yli/paper_data/viirs/. The AirNow observations can be downloaded from: https://files.airnowtech.org/?prefix=airnow/2020/. The CMAQ results can be downloaded from: http://air.csiss.gmu.edu/yli/paper_data/.

*Competing interests*. The authors declare that they have no conflict of interest.

*Author contributions*. YL and DT designed the study and performed the research with contributions from all co-authors. SF, RA, MS, and GG prepared plume rise code. SM, XZ, SK, and FL prepared fire emission data. RK prepared the MISR data and guided the evaluation of plume height estimation. YL and DT wrote and revised the paper, with input from YT, BB, and PC. All authors commented on drafts of the paper.

*Acknowledgments*. This work was financially supported by the NASA Health and Air Quality Program, NOAA Weather Program Office, and George Mason University College of Science. The observation data from NASA, NOAA, and US EPA are gratefully acknowledged.

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
