# Peer review of "Impacts of estimated plume rise on PM2.5 exceedance prediction during extreme wildfire events: A comparison of three schemes (Briggs, Freitas, and Sofiev)"

_EGUsphere, 2022_

## Author Comment (AC1)

*We thank the two reviewers for their insightful comments on our manuscript. Our responses to each of the reviewer's comments are provided below in italics.*

Reviewer #1:
The manuscript presents a study comparing three plume rise parameterizations on their ability to capture observed injection heights, and how they differ in their impacts on air quality and photochemistry on the short and long range. This study represents good contributions to the field and it's within the scope of ACP. I think the paper needs more work before it's ready for publication based on the comments below.

 My main comments are the following:
- The section evaluating injection heights is very short and could be greatly improved in many ways, including the addition of more cases that are more representative of the really extreme conditions that happened during this period. Also, the only injection height data used is that from MISR capturing fresh plumes. This does not capture the peak of fire activity that usually happens on the afternoon. For this I recommend including data from CALIPSO. While the chances for CALIPSO to capture fresh smoke are much lower, this event was so massive that most CALIPSO overpasses captured some section of the smoke emitted on a previous day. Thus an analysis could be done for the regional smoke heights rather than for smoke from individual fires, complementing that from MISR. The analysis should move a bit beyond a few study cases and try to capture the whole extend of the fire

*Thank you for the comments. We have compared the model simulations with the CALIPSO in our previous paper Li et al. (2021, GRL). The simulated vertical distribution of the smoke matched the observation near the wildfire source region and downwind area. In this paper, we have now added more comparisons with CALIPSO, in sections 2.4.1, 3.1, and relevant references.*

*In section 2.4.1, the following was added:*
*"CALIPSO is an Earth Science observation mission that was launched on 28 April 2006 and flies in a nominal orbital altitude of 705 km and an inclination of 98 degrees as part of a constellation of Earth-observing satellites. CALIPSO's lidar instrument, the Cloud-Aerosol Lidar with Orthogonal Polarization (CALIOP), provides high-resolution vertical profiles of aerosol and cloud attenuated backscatter signals at 532 nm and 1064 nm (Winker et al., 2007). The footprint of the lidar beam has a 100 m cross-section with an overpass around 1:30 p.m. local time. Level 1 attenuated backscatter profiles at 532 nm are used to calculate the plume height in this study following Amiridis et al. (2010)."*

*In Section 3.1, we added the discussion and a figure about CMAQ plume height compared to CALISPO:*
*"The vertical profiles of CMAQ simulated $PM_{2.5}$ are also compared to the CALIPSO aerosol vertical profile. The CALIPSO overpass occurs around 1:30 p.m. local time, closer to the peak fire behaviour in the afternoon than the MISR observations. The CALIPSO smoke injection heights are directly calculated from Level 1 attenuated backscatter profiles at 532 nm following Amiridis et al. (2010). There are several steps involved in this process. First, GBBEPx FRP data were used to locate the fire location along the CALIPSO swath. Then, a slope method (Pal et al., 1992) is applied to each profile to smooth out the original Level 1 532 nm attenuated backscatter*

*coefficient profiles at each fire point. Next, we calculate the steep gradient in the attenuated backscatter profiles. The height of the minimum gradient value is selected as the smoke injection height. Figure 4 shows the comparison between the CALIPSO plume height results and the estimated plume heights from the three plume rise schemes for west coast fires. In most cases, the plume heights from the three schemes are close to each other, especially for the cases with plume tops under 4 km. For strong fires with plume tops higher than 4 km, S12 seems to be more skillful than B69 and F07. It successfully simulates the high plume top observed by CALIPSO, whereas B69 and F07 tend to underestimate the plume heights.*

[Figure]

*Figure 4: Comparisons of plume top heights from three simulations: B69 (blue rectangle), F07 (orange dot), and S12 (red triangle), against aerosol height observations from the CALIPSO for west coast fires."*

*The following papers are added to the reference:*

*Amiridis, V., Giannakaki, E., Balis, D. S., Gerasopoulos, E., Pytharoulis, I., Zanis, P., Kazadzis, S., Melas, D., and Zerefos, C.: Smoke injection heights from agricultural burning in Eastern Europe as seen by CALIPSO, Atmos. Chem. Phys., 10, 11567–11576, https://doi.org/10.5194/acp-10-11567-2010, 2010.*

*Pal, S. R., Steinbrecht, W., and Carswell, A. I.: Automated method for lidar determination of cloud base height and vertical extent, Appl. Opt., 31, 1488–1494, 1992.*

*Winker, D. M., Hunt, W. H., and McGill, M. J. Initial performance assessment of CALIOP, Geophys. Res. Lett., 34, L19803, doi:10.1029/2007GL030135, 2007.*

- The article could be improved by being more thorough in it's literature review and using reference to better backup some statements. See some examples in the comments by line below

*Thanks for the comments. We have added these references (see the details in our responses to "Comments by line").*

- Some sections of the results were very qualitative on it's description, where I think a better job on being quantitative and using statistical metrics could have been done. More details in the comments below.

*Thank you. Please see the detailed responses below in the "Comments by line".*

Comments by line:

Intro. There have been multiple studies evaluating some of these plume injection height schemes beyond the Ye et al. (2021), so these previous findings need to be summarized. Some that come to mind can be found below, also look for work from Joe Wilkins. This literature review also can be used to motivate this study (i.e., what hasn't been done). Some uniqueness I see from these work include the comparison of these 3 schemes and the type of event studied (record-breaking wildfire season)

-Mallia, D., Kochanski, A., Urbanski, S. & Lin, J. Optimizing Smoke and Plume Rise Modeling Approaches at Local Scales. Atmosphere 9, 166 (2018).

- Wilmot, T. Y., Mallia, D. V., Hallar, A. G., and Lin, J. C.: Wildfire plumes in the Western US are reaching greater heights and injecting more aerosols aloft as wildfire activity intensifies, Scientific Reports, 12, 12400, 10.1038/s41598-022-16607-3, 2022.

-Sessions, W. R., Fuelberg, H. E., Kahn, R. A. & Winker, D. M. An investigation of methods for injecting emissions from boreal wildfires using WRF-Chem during ARCTAS. Atmos. Chem. Phys. 11, 5719–5744 (2011).

-Roy, A. et al. Effects of Biomass Burning Emissions on Air Quality Over the Continental USA: A Three-Year Comprehensive Evaluation Accounting for Sensitivities Due to Boundary Conditions and Plume Rise Height. in Energy, Environment, and Sustainability 245–278 (Springer Singapore, 2017). doi:10.1007/978-981-10-7332-8_12

*Thanks for the comments. All of the above papers are added to the paper. The following lines have been added to the paper:*

*"Wilmot et al. (2022) produced a decadal-scale wildfire plume rise climatology for the U.S. west coast and Canada and found trends toward enhanced plume heights, and the surface smoke injection to the free troposphere, which suggest a growing impact of wildfires on air quality and regional climate." added to line 54.*

*"Previous studies have found that the smoke injection height plays a vital role in smoke dispersion, as wind speed and direction generally vary with altitude (e.g., Mallia et al., 2018; Vernon et al., 2018;)." added to line 58.*

*"Sessions et al. (2011) tested the Freitas plume rise scheme with Weather Research and Forecasting and Chemistry (WRF-Chem) model and found that differences in injection heights produce different transport pathways. Roy et al. (2017) compared the simulated plume heights from two different approaches, Western Regional Air Partnership's (WRAP) plume model and*

*the Freitas plume model. Results show that the Freitas plume model got a better diurnal variation of the plume rise height. Mallia et al. (2018) tested different ways to distribute the fire emissions vertically for prescribed fires. Results indicated that plume height plays a critical role in determining how smoke distributes downwind of the fire." added to line 74.*

76. Add a reference to support these statements
*Thanks for the comment. The reference has been added.*

*"A series of large wildfires fuelled by accumulated biomass, heatwaves, and dry winds, burned more than 10 million acres (National Interagency Fire Center, 2020)."*

*National Interagency Fire Center. 2020 National Large Incident Year-to-Date Report (PDF). Geographic Area Coordination Center (Report). December 21, 2020.*

95-104. This paragraph is lacking any referencing, please add. Also, the notion that primary organic aerosol is also quite dynamic needs to be included as well.

*Thanks for the comment. The following papers have been added to section 2.1 as references:*

*Koppmann, R., von Czapiewski, K., and Reid, J. S.: A review of biomass burning emissions, part I: gaseous emissions of carbon monoxide, methane, volatile organic compounds, and nitrogen containing compounds, Atmos. Chem. Phys. Discuss., 5, 10455–10516, https://doi.org/10.5194/acpd-5-10455-2005, 2005.*

*Schlosser, J. S., Braun, R. A., Bradley, T., Dadashazar, H., MacDonald, A. B., Aldhaif, A. A., Aghdam, M. A., Mardi, A. H., Xian, P., and Sorooshian, A.: Analysis of aerosol composition data for western United States wildfires between 2005 and 2015: Dust emissions, chloride depletion, and most enhanced aerosol constituents, J. Geophys. Res.-Atmos, 122, 8951–8966, https://doi.org/10.1002/2017JD026547, 2017.*

*Seinfeld, J.H. and Pandis, S.N., Atmospheric Chemistry and Physics: From Air Pollution to Climate Change, 3rd edition, John Wiley & Sons, Hoboken, 2016.*

*In section 2.1 primary aerosol refer to the aerosols emitted from fires, and secondary aerosol refers to the aerosols generated after emitted. Line 117 has been changed to:*

*"Wildfire emissions include primary aerosols (direct emission) and large amounts of gases that can be oxidized to form secondary aerosols (generated after emission)."*

Section 2.2. A description of how model smoke injection height is derived is missing. Please be specific, e.g., what variable and what threshold is used, how is it mapped to the MISR pixels (location and time). Also include info on how AOD is derived and mapped to VIIRS.

*Thanks for the comments. More details of the three plume rise schemes are shown in section 2.3.*

*For B69 scheme, we added the equation:*

*"It uses a set of semi-empirical formulas to estimate plume injection height ($H_p$) in different atmospheric stability states (i.e., neutral, stable, and unstable). Heat flux (B), horizontal wind speed (U), static stability (S), and friction velocity ($x^*$) are used to estimate the plume injection height:*

$$H_p = \begin{cases} 1.33 \times BU^{-1}x^{*-2}, neutral \\ 2.6 \times (BU^{-1}S^{-1})^{\frac{1}{3}}, stable \\ 30 \times (BU^{-1})^{\frac{3}{5}}, unstable \end{cases} \qquad (1)$$

*Previous studies found that FRP is about 10-20% of the total fire heat (Wooster et al., 2005; Freeborn et al., 2008). In this study, we derive the heat flux from FRP provided by the GBBEPx dataset multiplied by a factor of 10 following Val Martin et al. (2012)."*

*For F07 we added more information:*
*"It takes in fire information, including fire area and heat flux, as well as atmospheric profile information, including temperature, moisture, density, and wind velocity. The plume top height is defined as the altitude at which the plume is neutrally buoyant, and is approximated as a vertical velocity < 1 m/s."*

*For S12 we added the equation:*
*"It utilizes FRP, PBL height ($H_{PBL}$), and the Brunt-Vaisala frequency in the free troposphere ($BV_{FT}$) to estimate the plume injection height from wildfires:*

$$H_p = \alpha H_{PBL} + \beta \left(\frac{FRP}{FRP_0}\right)^\gamma exp\left(-\frac{\delta BV_{FT}^2}{BV_0^2}\right) \qquad (2)$$

*Where $FRP_0$ is the reference fire radiative power which equals $10^6$ W, $BV_0$ is the reference Brunt-Vaisala frequency which equals $2.5\times10^{-4}$ $s^{-2}$, and α, β, γ, δ are constants."*

*The simulated plume height is mapped to MISR pixels using the nearest neighbor approach. The CMAQ model calculates AOD, which is compared to VIIRS AOD (Figure 6). We interpolated the model AOD to VIIRS pixels that passed quality control based on the nearest point method. The following lines are added to the paper:*

*"The simulated plume heights were interpolated to the MISR observation MISR pixels using the nearest neighbour approach." added to line 286.*

*"The CMAQ predicted AOD was interpolated to the VIIRS pixels that passed quality control using the nearest neighbor approach." added to line 412.*

Section 2.3. There are some details missing from the explanations of the injection schemes that could be useful to understand results. For instance, how is the plume distributed once the injection is computed? Is there a height of the bottom of the plume computed as well or how is it assumed? Is there a fraction of emission placed at the surface (so called "smoldering" emissions as stated in Freitas 2007)? If so, what % for each scheme? Due any of the parameterization consider different parameters for different fuels? Is the FRP used from GBBEPx a daily value? If so, is it applied as constant throughout the day or a given diurnal cycle is specified?

*Thanks for the comments. The plume is distributed between 0.5-1.5 times the plume injection height (default setting in CMAQ). In this study, we did not separate smoldering from flaming.*

*For the plume rise calculation, we did not consider different parameters for different fuels. But for emission and FRP, the GBBEPx applied different emission factors to different land use types (Zhang et al., 2012, 2019). The FRP used from GBBEPx is a daily value. When used in the model we gave the FRP a climatological diurnal cycle from the Western Regional Air Partnership (WRAP) work.*

*The following lines have been added to the paper in line 161:*
*"A climatological diurnal cycle from the Western Regional Air Partnership (WRAP) work was applied to the daily GBBEPx emission to derive hourly model-ready emission input."*

*The following lines have been added to the paper in line 236:*
*"After getting the estimated plume injection height from the three schemes, the fire emissions were distributed between 0.5-1.5 times plume injection height (default setting in CMAQ)."*

*The main focus of this study is to evaluate the impact of different plume injection heights on the near source and downwind air quality. For more information about the plume rise scheme or fire emission, please see the related references.*

161-162. Please expand a bit more on why the factor of 10 is applied.
*Thanks for the comments. The following lines and additional references have been added to the paper line 181:*

*"Previous studies found that FRP is about 10-20% of the total fire heat (Wooster et al., 2005; Freeborn et al., 2008). In this study, we derive the heat flux from FRP provided by the GBBEPx dataset multiplied by a factor of 10 following Val Martin et al. (2012)."*

167-168. Please add references for this sentence

*Thanks for the comments. We added the references here and in paper line 198:*

*"The Freitas scheme is the default plume rise scheme in WRF-Chem and has been widely used in many studies (e.g., Sessions et al., 2011; Val Martin et al., 2012; Roy et al., 2017; Mallia et al., 2018)."*

Section 2.4. VIIRS AOD data was not described. Please include any quality flags applied
*Thanks for the comments. The description for VIIRS AOD have been added to section 2.4.3:*

*"2.4.3 VIIRS AOD data*
*The simulated AOD results are compared to the VIIRS Enterprise AOD from Suomi-NPP. The VIIRS Enterprise Aerosol Algorithm retrieves AOD at the 750-m pixel level for the nominal wavelength of 550 nm using radiances from 11 VIIRS channels (412, 445, 488, 555, 672, 746, 865, 1240, 1378, 1610, and 2250 nm). The AOD is calculated separately for land and ocean using a lookup table of pre-computed values for several atmospheric parameters to simplify radiative transfer calculations."*

180. MISR injection heights have the limitation that MISR overpass is in the morning while peak fire behavior (and thus deeper injections) tends to be in the afternoon. This has been highlighted in previous work (paper below for instance). Since this is the only dataset used for evaluation in this work, this needs to be mentioned and taken into consideration when discussing results and deriving conclusions.

Kahn, R. A. et al. Wildfire smoke injection heights: Two perspectives from space. Geophys. Res. Lett. 35, (2008).

*Thanks for the comment. We add more comparison with CALIPSO which pass in the early afternoon at ~ 1:30 p.m. local time. The following discussion have been added to section 3.1:*

*"The vertical profiles of CMAQ simulated $PM_{2.5}$ are also compared to the CALIPSO aerosol vertical profile. The CALIPSO overpass occurs around 1:30 p.m. local time, closer to the peak fire behaviour in the afternoon than the MISR observations. The CALIPSO smoke injection heights are directly calculated from Level 1 attenuated backscatter profiles at 532 nm following Amiridis et al. (2010). There are several steps involved in this process. First, GBBEPx FRP data were used to locate the fire location along the CALIPSO swath. Then, a slope method (Pal et al., 1992) is applied to each profile to smooth out the original Level 1 532 nm attenuated backscatter coefficient profiles at each fire point. Next, we calculate the steep gradient in the attenuated backscatter profiles. The height of the minimum gradient value is selected as the smoke injection height. Figure 4 shows the comparison between the CALIPSO plume height results and the estimated plume heights from the three plume rise schemes for west coast fires. In most cases, the plume heights from the three schemes are close to each other, especially for the cases with plume tops under 4 km. For strong fires with plume tops higher than 4 km, S12 seems to be more skillful than B69 and F07. It successfully simulates the high plume top observed by CALIPSO, whereas B69 and F07 tend to underestimate the plume heights.*

[Figure]

*Figure 4: Comparisons of plume top heights from three simulations: B69 (blue rectangle), F07 (orange dot), and S12 (red triangle), against aerosol height observations from the CALIPSO for west coast fires."*

Figure 3. Having a visible image (MODIS Terra) including hotspots would help to visualize each scene.

*Thanks for the comment. MODIS Terra visible images have been added to Figure 3.*

[Figure]

*Figure 3: MISR plume heights superposed on the MODIS Terra visible images\* (a-d) and the comparisons of the observed plume height with the simulated plume heights (e-h) for the August 15th Milepost 21 fire easterly plume (a, e), westerly plume (b, f), August 31st Complex fire northerly plume (c, g), and southerly plume (d, h).*

*\*Source: MISR Active Aerosol Plume-Height (AAP) Project / R.A. Kahn, K.J. Noyes, J. Limbacher (NASA Goddard Space Flight Center), A. Nastan (JPL-Caltech), J. Tackett, J-P. Vernier (NASA Langley Research Center)*

Figure 3. It would also help to have the model boundary layer height as reference to assess boundary layer injections versus into the free-troposphere. Given the range shown by MISR, I would assume it's estimates contains a mixture of boundary layer smoke and injections. But the model doesn't show this variability, so it would be nice to understand why

*PBL heights have been added to Figure 3 (see above).*

Figure 3. Are these heights capturing mostly freshly emitted smoke or is here any recirculated smoke from the same or other fires?

*Thanks for the comment. They are freshly emitted smoke. More information about the MISR observation of the 2020 fires can be found here:*
*https://appliedsciences.nasa.gov/our-impact/news/nasa-researchers-analyze-properties-and-dispersion-smoke-plumes-august-complex*
*https://appliedsciences.nasa.gov/our-impact/news/nasa-researchers-analyze-properties-and-dispersion-smoke-plumes-august-2020*

*The main focus of this study is to evaluate the impact of plume injection heights on the near source and downwind air quality. More details about the MISR plume observation go beyond the scope of this paper.*

Figure 3. I find that evaluating only for 2 snapshots capturing 2 fires during August is a bit insufficient. There are likely many more opportunities during this 2 month period, especially during September where fires in the whole western US were exploding.

*Thanks for the comment. The MISR data is from MISR Active Aerosol Plume-Height (AAP) Project. We only found the information for two August fires. However, we added more comparisons and discussions with CALIPSO in section 3.1. See the response above.*

Figure 3. Also think in better ways of presenting the data, maybe aggregate MISR to the model resolution to avoid having so many repeated values for the model?

*Thanks for the comment. We think it is easier for the reader to understand if we aggregate the model results into the observation space (MISR grid).*

Figure 3. The model resolves the increase of height with distance, so you can do analysis to assess why is this happening. This is a bit counterintuitive as one would expect to earlier plume (i.e., further away) be emitted a lower altitudes. Or are the conditions such that the plume is just rising with time?

*Thanks for the comments. The main focus of this study is to evaluate the impact of plume injection heights on the near source and downwind air quality rather than the plume rise. This level of detail goes beyond the scope of this paper.*

191-202. Please be more quantitative. Higher by how much? Show some statistics

*Thanks for the comment. We have added more quantitative discussion:*

*"For the Milepost 21 Fire, the plume heights simulated by B69 and S12 are similar but 25% and 3% lower than that by F07 for the easterly and westerly plume. In the case of the August Complex Fire northerly plume, the plume heights simulated by S12 are 4% and 8% higher than that by B69 and F07, respectively. For the southerly plume, the plume heights simulated by B69 are 16% and 5% higher than that by F07 and S12."*

206-207 Maybe do statistical testing on the mean o backup this statement?

*Thanks for the comment. Box and whisker charts are added to Figure 3 to show more statistical results. Line 298 have been changed to:*

*"According to the box and whisker chart shown in the right panel of Figure 3, the simulated plume heights are all within the range of MISR observation. Overall, the simulated plume heights with all three schemes are reasonably comparable to the MISR observations."*

208-213. Need to better backup these statements. For instance, what were the stability conditions the model used for these fires. Also note Briggs not always shows higher injections.
*Thanks for the comment. We decide to delete this discussion.*

Figure 4. It would be nice to explore if these trends are also found on observations such at those from IMPROVE sites
*Thanks for the comment. We did not compare to IMPROVE sites for the following reasons. First, Figure 5 (original figure 4) shows the results of the column $PM_{2.5}$ rather than the surface $PM_{2.5}$ as we want to include the aerosol change both at the surface and in the plume aloft, while IMPROVE only measures the surface $PM_{2.5}$ concentration. Secondly, figure 5 shows the aerosol change due to the emission from biomass burning (the impact of other $PM_{2.5}$ sources was removed by subtracting the results of NoFire run). It is the difference between the two CMAQ simulations with and without fires, so we cannot compare it to the IMPROVE observation, which also include the $PM_{2.5}$ emitted from other sources. In addition, IMPROVE does not have observations for ammonium separately.*

Figure 5. It would be good to have a panel showing the average profile for one of the schemes to use it as a reference.
*Thanks for the comment. However, there is already a lot of information in Figure 6 (original figure 5). We don't want to make the figure too complicated. So, we did not add the average profile. But we add a difference ratio plot at the bottom (Figures 6 o and p), which can be used as a reference.*

Section 3.3. While VIIRS AOD is included, it doesn't seem to be used in the analysis. Satellite AOD tends to saturate around 5 so retrievals for the very fresh plumes are likely missing. If the model was not screened by these missing values then this likely explain why the model is overpredicting AOD on the locations of the fires. Ones you move away a bit from the fires the bias flip, with models tending to underpredict AOD. This discussion needs to happen before analysis is done comparing model runs.
*Thanks for the comment. We updated Figure 7 (original figure 6). When compared to the VIIRS, we applied the same saturation level, which is 5, to CMAQ results. When comparing the models, we removed the saturation level.*

*New Figure 7:*

[Figure]

*Figure 7: Two-month average AOD from VIIRS (a), B69 run (b), F07 run (c), and S12 run (d); the average AOD differences between F07 and B69 (e), and between S12 and B69 (f) from August 1st, 2020 to September 30th, 2020; and the average AOD difference ratio between F07 and B69 (g), and between S12 and B69 (h) during the same period.*

*The following lines have been added to line 411:*
*"Figure 7a shows the two-month averaged AOD from VIIRS compared with model simulations (Figures 7b-d). The CMAQ predicted AOD was interpolated to the VIIRS pixels that passed quality control using the nearest neighbor approach. When comparing the CMAQ AOD to VIIRS AOD*

*(Figures 7b-d), we applied VIIRS AOD saturation level (5) to CMAQ AOD results. In the west coast high peak region, all three runs capture the observed AOD high peak near the San Francisco region, but the simulated AOD peak is lower than VIIRS observed. The average AOD from VIIRS observation is higher than 2. However, among the three CMAQ runs, only F07 simulated an average AOD higher than 2. In the downwind region, all three CMAQ runs reproduce the general downwind transport pattern, but the simulated smoke affected region (AOD>0.5) is smaller than the observations.*

*.*
*Figures 7e-h show the AOD differences and the difference ratio (percentage of the difference relative to B69) between the different plume rise scheme simulations. When comparing different model simulations, the AOD saturation level is removed."*

295. Reference Figure 6 to make it clear you are comparing to that figure
*We deleted this discussion related to NO$_2$.*

310. Changes of 70% are described, but the color-scale of Fig 8 saturates at 30%. Exploring a scale that's not linear (like Fig 5) might work better.
*Thanks for the comment. The color-scale of Figure 9 (original figure 8) is changed:*

[Figure]

**Figure 9: Simulated and observed surface PM$_{2.5}$ from August 1$^{st}$, 2020, to September 30$^{th}$, 2020: a) average surface PM$_{2.5}$ simulated with B69 overlaid by AirNow observations; b) difference in averaged surface PM$_{2.5}$ between F07 and B69; c) difference between S12 and B69; d) and e) same as b) and c), but for the differences in percentage (%) between F07 and B69, and between S12 and B69, respectively.**

Figure 8. It would be nice to have a surface PM2.5 map for one of the simulations as a reference to better interpret the differences.
*Thanks for the comment. The surface PM$_{2.5}$ map is added to Figure 9 (see above).*

Figure 8. How much of these differences are due to differences in injection height versus assumptions of fraction of emissions placed at surface levels vs injected?

*All of the differences are due to the differences in injection heights. The only difference between the B69, F07, and S12 runs is the calculated fire injection heights. All three runs use the same emission and emission distribution algorithm.*

319-331. While 35 um/m3 is the standard, this divides the "Moderate" from "Unhealthy for sensitive groups" categories. However, it would be nice to see how the models in predicting the more extreme categories ("Unhealthy", "Very unhealthy", "Hazardous".) where more authorities might take more stringent measures

*Thanks for the comment. The results for "Unhealthy" and "Very unhealthy" are now included in Figure 10.*

[Figure]

*Figure 10: Predicted surface PM$_{2.5}$ concentrations above unhealthy levels by the S12, F07, and B69 runs for August 20$^{th}$, 2020: a) The daily mean surface PM$_{2.5}$ difference between F07 and B69 runs; b) simulated PM$_{2.5}$ exceedance regions by B69, F07, and S12 overlaid by AirNow observed exceedance (PM$_{2.5}$>35 μg/m$^3$); c) same to b) but for US EPA defined unhealthy regions (PM$_{2.5}$>55 μg/m$^3$); d) same to b) but for US EPA defined very unhealthy region (PM$_{2.5}$>150 μg/m$^3$). The brown color represents the region where the runs with all three schemes simulate PM$_{2.5}$ exceedances; the blue (red/yellow) color represents the region where only B69 (S12/F07) simulates the PM$_{2.5}$ exceedance; the green represents the region where both the B69 and F07 simulate the PM$_{2.5}$ exceedance; the magenta color represents the region where both the B69 and S12 simulate the PM$_{2.5}$ exceedance; the orange represents the region where both F07 and S12 simulate the PM$_{2.5}$ exceedance.*

*The discussion in line 564 have been changed to:*
*"The simulated PM$_{2.5}$ exceedance regions (PM$_{2.5}$>35 μg/m$^3$, defined by NAAQS, same level with US EPA defined unhealthy for sensitive groups), unhealthy region (PM$_{2.5}$>55 μg/m$^3$, defined by*

*US EPA), and very unhealthy region (PM$_{2.5}$>150 µg/m$^3$, defined by US EPA) by different plume rise schemes overlaid by AirNow observed exceedance for the same day are shown in Figures 10b, c, and d. According to Figure 10b and c, on August 20$^{th}$, 2020, B69 and S12 simulated more PM$_{2.5}$ exceedance and a larger unhealthy region in the downwind regions (Wyoming (WY) and Idaho (ID), magenta and blue region), whereas F07 and S12 simulated more exceedance and a larger unhealthy region in the southeastern U.S. (yellow and orange region), where prescribed fires were the major biomass burning sources. In WY and ID, where F07 did not simulate the PM$_{2.5}$ exceedance whereas B69 and S12 did, the difference between F07 and B69 reached 15 µg/m$^3$ (Figure 10a). Furthermore, B69 and S12 simulate some very unhealthy regions in Nevada, whereas F07 simulates more very unhealthy regions in central and south California.''*

319-331. This analysis is based on one day. A way to generalize this analysis could be to show a map or difference maps of the number of days the models predict exceedances.

*Thanks for the comment. We have added a new figure (Figure 11) to show the total number of exceedance days and the difference in the number of exceedances between different:*

[Figure]

*Figure 11: The CMAQ B69 predicted total number of PM$_{2.5}$ exceedance days during Aug-Sep, 2020 (a); the difference in the number of predicted exceedance days between B69 and F07 (b), and between B69 and S12 (c).*

*The following discussions have been added to the paper in line 634:*

*"The total number of predicted exceedance days from the B69 simulation and the differences between B69, F07, and S12 are shown in Figure 11. All the states in the western coast and mountain region experienced at least one day of $PM_{2.5}$ exceedance (Figure 11a). Most of the region in northern California experienced more than 20 exceedance days, with a maximum of more than 35 days. F07 and S12 simulate more exceedance days on the west coast near the source region and in the southeast. The difference in the exceedance days could be as large as 20 days in northern California. B69 simulates more exceedance days in downwind regions such as Nevada, Idaho, Montana, and Wyoming. The difference could reach 4 days in the downwind regions."*

Minor Edits

191: ACP won't allow links, convert it to a reference
*Thanks for the comment. The link is deleted and added to the data availability part.*

321-325. A lot of this text is already in Fig 9 caption so no need to repeat.
*Thanks for the comment. They are deleted.*

Reviewer #2:

This paper used an offline CMAQ model with three different plume rise schemes to discuss the impact of injection plume height on local and downwind aerosols' chemical composition as well as the photochemical processes. The paper clearly explains the basic settings of the CMAQ model and the different input parameters in three plume models.

This work has compared the predicted AOD among three plume models to illustrate their different prediction performance in the source and downwind area, while a detailed comparison between VIIRS observed AOD and modeled AOD is expected to further validate the model prediction accuracy. This work has made the comparison of PM exceedance regions between the model prediction and the AirNow observation on Aug 20, 2020. The result shows good consistency.

The analysis of the plume models' impacts on photochemistry mainly focuses on the photolysis of NO2. We expect more observation evidence on NO2concentration to validate the model prediction. The relationship between NO2 concentration or photolysis rate and other species concentration which have adverse effects on human health (e.g., ozone) needs to be further established.

General comments:
Introduction:
Authors explained the different parameters used in three schemes for plume height estimates. We expect the authors to explain how the later-published schemes of plume height simulation improve the modeling accuracy, in general. Also, what is the limitation of each model.
*Thanks for the comment. More details of the three plume rise schemes are shown in section 2.3.*

*For B69 scheme, we add the equation:*
*"It uses a set of semi-empirical formulas to estimate plume injection height ($H_p$) in different atmospheric stability states (i.e., neutral, stable, and unstable). Heat flux (B), horizontal wind speed (U), static stability (S), and friction velocity ($x^*$) are used to estimate the plume injection height:*

$$H_p = \begin{cases} 1.33 \times BU^{-1}x^{*-2}, neutral \\ 2.6 \times (BU^{-1}S^{-1})^{\frac{1}{3}}, stable \\ 30 \times (BU^{-1})^{\frac{3}{5}}, unstable \end{cases} \qquad (1)$$

*Previous studies found that FRP is about 10-20% of the total fire heat (Wooster et al., 2005; Freeborn et al., 2008). In this study, we derive the heat flux from FRP provided by the GBBEPx dataset multiplied by a factor of 10 following Val Martin et al. (2012). "*

*For F07 we add more information:*
*"It takes in fire information, including fire area and heat flux, as well as atmospheric profile information, including temperature, moisture, density, and wind velocity. The plume top height is defined as the altitude at which the plume is neutrally buoyant, and is approximated as a vertical velocity < 1 m/s."*

*For S12 we add the equation:*
*"It utilizes FRP, PBL height ($H_{PBL}$), and the Brunt-Vaisala frequency in the free troposphere ($BV_{FT}$) to estimate the plume injection height from wildfires:*

$$H_p = \alpha H_{PBL} + \beta \left(\frac{FRP}{FRP_0}\right)^\gamma exp\left(-\frac{\delta BV_{FT}^2}{BV_0^2}\right) \qquad (2)$$

*Where $FRP_0$ is the reference fire radiative power which equals $10^6$ W, $BV_0$ is the reference Brunt-Vaisala frequency which equals $2.5 \times 10^{-4}$ $s^{-2}$, and $\alpha$, $\beta$, $\gamma$, $\delta$ are constants."*

*All three schemes are widely used schemes, but using different methods to calculate the plume height. We don't want to say which one is the best, or which one is the worst in this paper. Also, the main focus is the plume height impact on downwind air quality, not plume rise schemes. More details about these schemes are beyond the scope of this paper.*

Method:
The CMAQ model domain has a spatial resolution of 12 km × 12 km. The scale of an wildfires in the western US is normally smaller than the spatial resolution of the defined CMAQ domain (Biomass burning emission is a 0.1 degree product). Please provide the dimension information (or related inforation) of the studied fire to explain the choice of domain resolution.
*Thanks for the comment. We use the same horizontal and vertical resolution as NOAA's operational National Air Quality Forecasting Capability (NAQFC). The results of this paper and the following studies will be used to improve NAQFC. The following lines have been added to line 151:*

*"(the same horizontal and vertical resolution as NOAA's operational National Air Quality Forecasting Capability)"*

In section 2.1 Experiment Design, authors have mentioned the reaction pathway from VOCs to SOA. However, in the result analysis part, section 3.2, the contribution of SOA in the total OM has not been mentioned. The potion of OM in the total PM2.5 seems to be completely regarded as the primary emission. Combining primary OA and SOA together may introduce errors in the further discussion on particle/gas transport issue.
*This is a good point and we have clarified it in Section 3.2 (first paragraph):*
*"In this section, we investigate the impact of plume injection height on different $PM_{2.5}$ chemical components. Figure 5 shows the distribution of simulated $PM_{2.5}$ components from both direct emissions and secondary formation (the impact of other $PM_{2.5}$ sources was removed by subtracting the results of NoFire run) from B69, F07, and S12 in three different regions."*

*Primary and secondary aerosols have different sources (primary aerosols refer to the aerosols emitted by the fire, secondary aerosols refer to the aerosol generated after emitted), but could be the same species, just like the organic carbon, sulfate, ammonium, and nitrate in figure 2. In CMAQV5.3.1, primary and secondary aerosols are blended to do the reaction and dispersion. It is hard to separate secondary aerosols from primary aerosols in the output. However, we don't think it will affect our discussion, as the main focus is the overall impact (combining primary and secondary aerosols) of plume rise on air quality.*

Results:
Figure 5 plots the specific chemical component of PM2.5 against the distance (km) from the source point. Is this distance along certain smoke transport pathway? If so, which specific pathway you chose to sample the modeled concentration of different species.

*Thanks for the comment. Figure 5 (Now it is Figure 6) shows the zonal mean results (average for each latitude), not along certain smoke transport pathways.*
*Line 378 has been changed to clarify it:*

*"Figure 6 shows the difference in the zonal mean (average for each latitude) concentrations of six major PM$_{2.5}$ species"*

Specific comments:
Line 48: PM2.5 definition: Particle's aerodynamic diameter is less than 2.5 μm
*Thanks for the comment. Line 49 has been changed to:*

*"O'Neil et al. (2021) discuss the regional health impacts of the 2017 Northern California wildfires and estimated 83 excess deaths from exposure to PM$_{2.5}$ (i.e., particles having aerodynamic diameter less than 2.5 μm), of which 47% of the deaths were attributable to wildfire smoke during the smoke episode."*

Line 48: "47%" in mass or other types of measure?
*Thanks for the comment. It is 47% of the estimated excess deaths. Line 49 has been changed to:*

*"O'Neil et al. (2021) discuss the regional health impacts of the 2017 Northern California wildfires and estimated 83 excess deaths from exposure to PM$_{2.5}$ (i.e., particles having aerodynamic diameter is less than 2.5 μm), of which 47% of the deaths were attributable to wildfire smoke during the smoke episode."*

Line 62: "Irregular large point sources". What does this terminology mean? The boundary of the source is irregular? Then why is a point source?
*Thanks for the comment. In CMAQ, fire is treated as an irregular point source emission, as the occurrence and location of the emission are not regular or fixed. Line 66 have been changed to make it clear:*

*"...the Briggs scheme was not designed for irregular occurrence large point source emissions, such as forest fires..."*

Line 78: the unit of "3720". Daily, hourly cases?
*Thanks for the comment. The NAAQS defined PM$_{2.5}$ exceedance based on daily PM$_{2.5}$. Line 92 has been changed to make it clear:*

*"From late August to early October 2020, the West Coast wildfires contributed 23% of surface PM$_{2.5}$ pollution nationwide and caused 3,720 observed PM$_{2.5}$ exceedances (daily PM$_{2.5}$>35μg/m$^3$ based on National Ambient Air Quality Standards; Li et al., 2021). "*

Line 156: What's the result of this reason?
*Thanks for the comment. Line 244 has been changed to:*

*"The main focus of this study is to evaluate the impact of different plume injection heights on the near source and downwind air quality, and the two-month average state is more important to our results and future health studies."*

Line 209: Define "ABL" before using it
*This paragraph is deleted.*

Line 225: Is "OM" here the same as organic carbon you defined in section 2.1, which only refers to primary organic carbon?
*Thanks for the comments. In section 2.1 primary aerosol refer to the aerosols emitted from fires, and secondary aerosol refers to the aerosols generated after emitted. They can be the same species. As in Figure 2, there are Sulfate, Nitrate, and Ammonium in both primary and secondary aerosols (same as the organic carbons). Primary and secondary aerosols are not different species, but different sources. In section 2.1, organic carbon does not only refer to primary organic carbon. OM is organic matter, it is different from organic carbon in that it includes all the elements (hydrogen, oxygen, nitrogen, etc) that are components of organic compounds, not just carbon. OC is included in OM. In CMAQ, OM is used to calculate the total PM$_{2.5}$. Therefore, we use OM instead of OC in the discussion in section 3.*

*We added a note to Figure 2 to make it clear:*
*"$^+$Note: The CMAQ model separates organic carbon (OC) and non-carbon elements (O, H, etc) in organic matter (OM)."*

Line 226: Clarify that the composition of PM2.5 in this section is surface PM2.5, or PM2.5 under PBL, or column PM2.5. Line 232 mentioned "surface PM2.5", and the conclusion of this section is "integrated over all vertical layer".
*Thanks for the comment. Sorry, it is a typo. Not surface PM$_{2.5}$. It should be PM$_{2.5}$ integrated over all vertical layers. The word "surface" is deleted.*

Figure 4: The negative sign before the longitude is unnecessary
*Thanks for the comment. We decide to keep it.*

Line 249: Unify the representation of longitude: either 115° W or -115° throughout the paper
*Thanks for the comment. We have changed it to "in the west of 115° W".*

Figure 6: The comparisons between VIIRS AOD and modeled results may be needed to demonstrate the prediction accuracy of different plume models.
*Thanks for the comments. More discussions about the comparisons between VIIRS AOD and model results have been added to the paper section 3.3:*

*"Figure 7a shows the two-month averaged AOD from VIIRS compared with model simulations (Figures 7b-d). The CMAQ predicted AOD was interpolated to the VIIRS pixels that passed quality control using the nearest neighbor approach. When comparing the CMAQ AOD to VIIRS AOD (Figures 7b-d), we applied VIIRS AOD saturation level (5) to CMAQ AOD results. In the west coast high peak region, all three runs capture the observed AOD high peak near the San Francisco region, but the simulated AOD peak is lower than VIIRS observed. The average AOD*

*from VIIRS observation is higher than 2. However, among the three CMAQ runs, only F07 simulated an average AOD higher than 2. In the downwind region, all three CMAQ runs reproduce the general downwind transport pattern, but the simulated smoke affected region (AOD>0.5) is smaller than the observations."*

Line 286: Thicker smoke in this study doesn't necessarily mean higher AOD. Thicker smoke somehow may be attributed to a diluted plume because of the different plume height modeled by different schemes. A basic assumption in this study is the primary biomass burning emissions among three models are similar (or identical).
*Here thicker smoke refers to a higher concentration of total column $PM_{2.5}$. Aerosol optical depth (AOD) is a quantitative estimate of the amount of aerosol present in the atmosphere. Since all three runs use the same emission, the aerosol type should be similar, a higher AOD can indicate higher smoke concentrations.*

Typos or other improvement suggestions:
Line 25: lower case "western"
*We decided to keep "Western United States". It's a proper noun. Proper nouns should be capitalized.*

Line 46: two "annual"
*Thanks. Line 47 has been changed to:*
*"The global average mortality attributable to landscape fire smoke exposure was estimated at 339,000 deaths annually (Johnston et al., 2012)."*

Line 63: Start a new sentence to declare the second limitation of Briggs scheme.
*Thanks for the comment. It has been changed to:*
*"Also, some of the input parameters, such as heat flux, are difficult to obtain."*

Line 69: Please provide the reference of Siberia study
*Thanks for the comment. The data are from the MISR plume height project. The following two papers have been added as the reference:*
*Line 72 is changed to:*

*"The parameters of the new scheme were determined using the plume height observations collected by the Multi-angle Imaging SpectroRadiometer (MISR) Plume Height Project (Kahn et al., 2008; Mazzoni et al., 2007) in North America (Val Martin et al., 2010) and Siberia."*

*The following lines are added to the reference part:*
*Kahn, R. A., Chen, Y., Nelson, D. L., Leung, F.-Y., Li, Q., Diner, D. J., and Logan, J. A. Wildfire smoke injection heights: Two perspectives from space, Geophys. Res. Lett., 35, L04809, doi:10.1029/2007GL032165, 2008*

*Mazzoni, D., Logan, J.A., Diner, D., Kahn, R. A., Tong, L., and Li, Q.: A data-mining approach to associating MISR smoke plume heights with MODIS fire measurements, Remote Sens. Environ., 107, 138-148, 2007.*

Line 69: "height"
*Thanks. It's a typo. We have changed it to "height".*

Line 72: Suggest start a new sentence here.
*Line 87 has been changed to:*
*"They found that there was a large spread of the modelled plume heights."*
Line 162: "from… from…" redundancy
*Line 190 has been changed to:*
*"...we derive the heat flux from FRP provided by the GBBEPx dataset ..."*

Line 185: lower case
*We think it is okay to use "the Equator" here. The Equator is the invisible line that runs around the center of the Earth at 0 degrees latitude. An equator is an imaginary line around the middle of a planet or other celestial body. It's a proper noun. Proper nouns should be capitalized.*

Line 263: lower case
*We think it is okay to use "the Earth", as we're referring to the specific planet. The word "earth" refers to the soil or surface of a stratum. Earth is a proper noun. It names a specific place. Proper nouns should be capitalized.*

Figure 6: Increase the tick font size of the colorbar
*Thanks for the comment. The font size has been enlarged in Figure 6 (Now it is figure 7).*

Line 296: The difference ratios in NO2 is higher than the ones of AOD can only prove the concentration of NO2 is not linearly proportional to AOD.
*Thanks for the comment. The discussion of $NO_2$ is deleted. The change of $NO_2$ is very complicated and beyond the scope of this paper.*

Line 297: The reaction rate of NO2 (for this NO2 -> NO + O reaction) is the product of <NO2_IUPAC10> and the concentration of NO2. The comparison of reaction rates between different plume schemes is needed to support your conclusion.
*Thanks for the comment. The discussion of $NO_2$ is deleted.*

Figure 7: Increase the tick font size of the colorbar
*Thanks for the comment. The font size has been enlarged in Figure 7 (Now it is figure 8).*

Other comments
Line 67: This sentence means Sofiev scheme used the MISR observed plume height to determine the modeled plume height. I am a bit confused about it (Line 67 - 69).
*The Sofiev et al. (2012) scheme uses the plume height collected within the MISR Plume Height Project to calculate the parameters used in the Sofiev et al. (2012) scheme. More information can be found in the reference:*

*Sofiev, M., Ermakova, T., and Vankevich, R.: Evaluation of the smoke-injection height from wild-land fires using remote-sensing data, Atmos. Chem. Phys., 12, 1995–2006, https://doi.org/10.5194/acp-12-1995-2012, 2012.*

Line 76: Please provide the reference of the burned area. (I remembered the burned area in the western US in 2020 is high but below 10 million acres. The entire US is larger than 10 million acres)

*Thanks for the comment. The reference has been added.*
*National Interagency Fire Center. 2020 National Large Incident Year-to-Date Report (PDF). Geographic Area Coordination Center (Report). December 21, 2020.*

Reviewer #3
Review comment on "Impacts of estimated plume rise on PM2.5 exceedance prediction during extreme wildfire events: A comparison of three schemes (Briggs, Freitas, and Sofiev)

The authors compared three popular plume rise schemes, namely Briggs 1969, Freitas 2007 and Sofiev 2012, and their impacts on the simulated plume heights, AOD, PM2.5 and NO2 photochemistry using the CMAQ model driven by WRF meteorology data for the 2020 western U.S. wildfire season. With global warming, the increasing trend in western U.S. fire activities, and the need to predict hazardous air quality associated with wildfires, the study would make a timely and significant contribution to wildfire and air quality modeling science. So publication is recommended. However, I believe, the presentation can be significantly improved to increase the scientific impact of this study.

Major comments:
The descriptions of each of the plume rise schemes are short. More details of the schemes could be provided to help readers know better of the differences of the schemes (length of description can be doubled or tripled). Also, the authors focus mainly on plume top height, however plume extension (top and bottom of a plume) in the vertical at emission is as important as plume top. Information about plume vertical extension at emission and how emission mass is distributed in the vertical (e.g. evenly or weighted) from the schemes should be provided.

*Thanks for the comments. More details of the three plume rise schemes have been added in section 2.3.*

*For B69 scheme, we add the equation:*
*"It uses a set of semi-empirical formulas to estimate plume injection height ($H_p$) in different atmospheric stability states (i.e., neutral, stable, and unstable). Heat flux (B), horizontal wind speed (U), static stability (S), and friction velocity ($x^*$) are used to estimate the plume injection height:*

$$H_p = \begin{cases} 1.33 \times BU^{-1}x^{*-2}, neutral \\ 2.6 \times (BU^{-1}S^{-1})^{\frac{1}{3}}, stable \\ 30 \times (BU^{-1})^{\frac{3}{5}}, unstable \end{cases} \quad (1)$$

*Previous studies found that FRP is about 10-20% of the total fire heat (Wooster et al., 2005; Freeborn et al., 2008). In this study, we derive the heat flux from FRP provided by the GBBEPx dataset multiplied by a factor of 10 following Val Martin et al. (2012)."*

*For F07 we add more information:*

*"It takes in fire information, including fire area and heat flux, as well as atmospheric profile information, including temperature, moisture, density, and wind velocity. The plume top height is defined as the altitude at which the plume is neutrally buoyant, and is approximated as a vertical velocity < 1 m/s."*

*For S12 we add the equation:*
*"It utilizes FRP, PBL height ($H_{PBL}$), and the Brunt-Vaisala frequency in the free troposphere ($BV_{FT}$) to estimate the plume injection height from wildfires:*

$$H_p = \alpha H_{PBL} + \beta \left(\frac{FRP}{FRP_0}\right)^\gamma \exp\left(-\frac{\delta BV_{FT}^2}{BV_0^2}\right) \qquad (2)$$

*Where $FRP_0$ is the reference fire radiative power which equals $10^6$ W, $BV_0$ is the reference Brunt-Vaisala frequency which equals $2.5\times10^{-4}$ $s^{-2}$, and $\alpha$, $\beta$, $\gamma$, $\delta$ are constants."*

*The plume is distributed between 0.5-1.5 times the plume injection height (default setting in CMAQ).*

*The following lines have been added to the paper in line 236:*
*"After getting the estimated plume injection height from the three schemes, the fire emissions were distributed between 0.5-1.5 times plume injection height (default setting in CMAQ)."*

It's not clear how model and MISR plume heights were compared. The model and MISR observations don't have the same spatial and temporal resolutions, and MISR observations are not continuous in time and space. So some spatiotemporal interpolation is expected. The treatment of the model and MISR data for comparison should be clearly stated.
*Thanks for the comment. The simulated plume height is mapped to MISR pixels based on the nearest point. We compared the CMAQ with MISR at the MISR local overpass time of around 19 UTC. local time. The following lines have been added to line 284.*

*"The simulated plume heights from three simulations: B69, F07, and S12 are compared with MISR observations for the Milepost 21 Fire on August 15th, 2020, and the August Complex Fire on August 31st, 2020 (Figure 3) at the MISR local overpass time of around 19 UTC. The simulated plume heights were interpolated to the MISR observation MISR pixels using the nearest neighbor approach."*

Section 3.3 and conclusion: Why do F07 and S12, which tend to have lower plume height than B69 near source region, have higher AOD than B69 near source region? What is the column total PM2.5 differences in the source region for the three schemes? Is the difference small? (You could consider providing total/average PM2.5 or even better dry mass in Figure 4 for the different regions and schemes) If so, what causes the 20-30% AOD difference in source region? Is it purely because of different vertical distributions of same mass? For example, there could be more aerosols in the lower altitude in F07 and S12 (and RH tends to be higher than higher altitude), so that hygroscopic growth of smoke in the lower layer leads to the higher AOD? Or is it due to different SOA production rate? The authors should be able to provide some discussions through analysis.

*Thanks for the comment. Previous studies found that a higher plume height will inject more aerosol into the free troposphere where the wind speed is stronger and accelerate the dispersion*

*of the fire pollution. Therefore, the higher plume height will lead to lower AOD near the source region but higher AOD in the downwind region. We have added more discussion in section 3.3 to make it clear:*

*"One possible reason that B69 predicts lower AOD near the source region and higher AOD in the downwind region compared to F07 and S12 is that a higher plume height will inject more aerosol into the free troposphere where the wind speed is stronger, accelerating the dispersion of the fire pollution. Therefore, the higher plume height will lead to lower AOD near the source region but higher AOD in the downwind region. The result is consistent with previous studies (Mallia et al., 2018; Vernon et al., 2018; Li et al., 2020)."*

*The AOD is a quantitative estimate of the amount of aerosol present in the atmosphere. The difference in the total column $PM_{2.5}$ could be informed by the AOD comparison. Therefore, we think the plots for total column $PM_{2.5}$ is not necessary.*

*The only difference between the three runs is the estimated plume injection height. We use the same chemistry mechanism, same emission, and same dynamic for the three runs. Any difference between the three runs is caused by the difference in the plume injection height. Of course, the difference in the plume height will lead to the difference in dispersion, and aerosol concentration, and affect the reaction rate of some chemistry or photolysis reactions which we discussed in the later part of section 3.3. However, all of the following differences are the results of the difference in the plume injection height. We have added more discussion in section 3.3 to make it clear:*

*"The difference in the dispersion of fire pollution caused by the various estimated plume injection heights leads to further differences in the chemistry and photolysis reactions. Previous studies found that the thicker smoke, indicated by higher AOD, may absorb and/or scatter a larger fraction of sunlight, hence affecting photolysis reactions (Dickerson et al., 1997; Castro et al., 2001; Kumar et al., 2014; Baylon et al., 2018)."*

*Since the difference in the photolysis rate is not the main focus of this study. We think more detailed discussions are beyond the scope of this paper.*

Figure 9: This is a case study of PM2.5 exceedance. Using a color wheel with overlapping colors to represent simulated PM2.5 exceedance regions from the three schemes is brilliant. I do have a few questions though: Why August 20th is chosen as the case? The authors should provide a reason. Since this is a case study, some background of the wild fires and PM transport should be provided. Did you compare the plume heights with MISR (This case was not included in the earlier section or Figure 3)? Why F07-B69 difference in daily surface PM2.5 is provided, but not S12-B69? The authors should provide the reasoning of leaving this comparison out or making this comparison.

*Thanks for the comments. Based on the previous discussion, the difference between F07 and B69 is larger than S12 and B69, so we just show the plot for F07-B69 to save space. August 20 is the first peak of the 2020 wildfire season. There is no MISR observation in the fire source region on that day. We have added more explanation in the paper:*
*"...for Aug $20^{th}$, 2020 (the first fire peak during the study period)."*

*Also, we add more discussion and a figure to show the map of the total number of PM$_{2.5}$ exceedance days and the difference in the total number of exceedance days between the three runs. This analysis can show the overall condition of the whole study period.*
*The following lines and figure 11 have added to the end of section 3.4:*

*"The total number of predicted exceedance days from the B69 simulation and the differences between B69, F07, and S12 are shown in Figure 11. All the states in the western coast and mountain region experienced at least one day of PM$_{2.5}$ exceedance (Figure 11a). Most of the region in northern California experienced more than 20 exceedance days, with a maximum of more than 35 days. F07 and S12 simulate more exceedance days on the west coast near the source region and in the southeast. The difference in the exceedance days could be as large as 20 days in northern California. B69 simulates more exceedance days in downwind regions such as Nevada, Idaho, Montana, and Wyoming. The difference could reach 4 days in the downwind regions."*

[Figure]

*Figure 11: The CMAQ B69 predicted total number of PM$_{2.5}$ exceedance days during Aug-Sep, 2020 (a); the difference in the number of predicted exceedance days between B69 and F07 (b), and between B69 and S12 (c).*

Minor comments:
Line 69: "heigh" should be "height".
*Thanks. The "heigh" is changed to "height".*

Line 77-78: Please define "PM2.5 exceedance". Is it based on daily-mean or hourly PM data? Also for the 3720 observations, how many sites are the observations based on?
*Thanks for the comments. The definition of NAAQS $PM_{2.5}$ exceedance has been added: "...observed $PM_{2.5}$ exceedances (daily $PM_{2.5}>35\,\mu g/m^3$ based on National Ambient Air Quality Standards..."*
*The number of AirNow $PM_{2.5}$ is not fixed every day. Usually, it is around 1000. For more details about the previous study, please refer to the following paper:*
*Li, Y., Tong, D., Ma, S., Zhang, X., Kondragunta, S., Li, F., & Saylor, R.: Dominance of wildfires impact on air quality exceedances during the 2020 record-breaking wildfire season in the United States. Geophysical Research Letters, 48, e2021GL094908. https://doi.org/10.1029/2021GL094908, 2021.*

Line 79-80: There is no direct visual link between "hazy" and AOD shown in Figure 1. I would suggest adding a matching VIIRS true color image and/or define "hazy" in terms of AOD and PM values.
*Thanks for the comment. Figure 1 is changed. VIIRS true color image is added to figure 1.*

**a) 20200915 VIIRS True Color Image**

[Figure]

**b) 20200915 VIIRS 550 nm AOD**

*Figure 1. Observations of wildfire smoke on September 15, 2020, over the continental United States by the Visible Infrared Imaging Radiometer Suite (VIIRS) aboard the Suomi-NPP satellite: a) true color image and b) 550 nm aerosol optical depth (AOD).*

*The following sentence has been changed to make it clear in line 96:*
*"caused hazy days (indicated by the high AOD region) in 19 states (Figure 1)."*

Figure 2: What is the data source of this figure and pie chart in the figure. The authors should cite some papers on fire emission chemistry (currently there are none) in the introductory paragraph (line 95-105) of experiment design. Below are a few examples:
Koppmann, R., von Czapiewski, K., and Reid, J. S.: A review of biomass burning emissions, part I: gaseous emissions of carbon monoxide, methane, volatile organic compounds, and nitrogen containing compounds, Atmos. Chem. Phys. Discuss., 5, 10455–10516, https://doi.org/10.5194/acpd-5-10455-2005, 2005.
Schlosser, J. S., Braun, R. A., Bradley, T., Dadashazar, H., MacDonald, A. B., Aldhaif, A. A., Aghdam, M. A., Mardi, A. H., Xian, P., and Sorooshian, A.: Analysis of aerosol composition data for western United States wildfires between 2005 and 2015: Dust emissions, chloride depletion, and most enhanced aerosol constituents, J. Geophys. Res.-Atmos, 122, 8951–8966, https://doi.org/10.1002/2017JD026547, 2017.

*Thank you for providing us with these papers. All are added to the main text and the reference list*.

Line 145-146: To be more accurate, the vertical profiles of PM2.5 would match the "vertical profiles of backscatter" from CALIPSO.
*Thanks for the comment. Line 177 has been changed to:*
*"the simulated $PM_{2.5}$ vertical profiles in the West Coast and Central U.S. matched the vertical profiles of backscatter from Cloud-Aerosol Lidar and Infrared Pathfinder Satellite Observations (CALIPSO) ..."*

Line 156-157: Not a complete sentence.
*Thanks for the comment. Line 244 has been changed to:*
*"The main focus of this study is to evaluate the impact of different plume injection heights on the near source and downwind air quality, and the two-month average state is more important to our results and future health studies."*

Line 183: "70°" instead of "70".
*Thanks for the comment. "70" has been changed to "70°".*

Line 232: "….which include nitrate formation from both wildfires and anthropogenic emission". This is confusing. I would expect no anthropogenic influence , as "the impact of other PM2.5 sources was removed by subtracting the results of NoFire run" from line 224.
*Thanks for the comment. The part about "anthropogenic emission" is removed. Line 354 has been changed to:*
*"A higher portion of $NO_3$ in the downwind region than in the source region reflects the decreased contribution of primary aerosols, and increases in secondary aerosols."*

It may be worth labeling the longitudes on the upper x axis on the geographical plots (e.g. at least Figure 1), so that readers would know the projection of the maps and where the division lines lie between the regions. This information is currently not straight forward. An alternative is to plot the division lines on the maps.
*Thanks for the comment. The longitudes on the upper x-axis are added to all maps.*

Line 298-299: "The consumption of NO2 is slowed down so that the NO2 concentration is higher in the high AOD area." I think you meant "so" instead of "so that".
*The discussion related to $NO_2$ is deleted.*

Line 271: You could consider updating the subsection title to include the impact of plume rise on photochemistry besides AOD.
*Thank you for the comment. The subsection title has been updated:*
*"3.3 Impact of estimated plume rise on aerosol optical depth and photochemistry"*

Line 321-325: The description of color scales is already included in the figure caption, which is the right place. It is redundant here in the text.
*Thanks for the comment. They are deleted in the text.*

Figure 8: It would be helpful if a plot of the average surface PM2.5 from B69 overlaid with AirNow measurement is provided, as the difference plots here are based on B69 surface PM2.5. Also the addition of AirNow would provide some kind of evaluation for the model.

*Thanks for the comment. We added the plot of the average surface PM$_{2.5}$ from B69 overlaid with AirNow measurements to Figure 9 (old Figure 8):*

[Figure]

*Figure 9: Simulated and observed surface PM$_{2.5}$ from August 1$^{st}$, 2020, to September 30$^{th}$, 2020: a) average surface PM$_{2.5}$ simulated with B69 overlaid by AirNow observations; b) difference in averaged surface PM$_{2.5}$ between F07 and B69; c) difference between S12 and B69; d) and e) same as b) and c), but for the differences in percentage (%) between F07 and B69, and between S12 and B69, respectively.*

---

## Referee Report (RR1)

The revised manuscript has addressed the reviewers' comments and provided additional analysis to improve the demonstration. Further comments and discussion to the revised manuscript is provided below:

**Minor revision suggestions:**

*Fig. 6*

Fig 6 shows the zonal averaged concentrations throughout the two-month period to illustrate how different plume rise models affect the spatial and vertical distribution of the studied atmospheric species, and species' fractional contribution in PM2.5.

The occurrence of wildfires in the US is quite sporadic. The zonal averaging is appropriate to highlight the dominant pattern of the fire activities in the US. What is the latitude range you've applied for the zonal averaging? Are you using a constant averaging range (from the west to the east) or a changing averaging range as the smoke dispersed?

*Result analysis of Fig. 6 (Section 3.3)*

Authors explained the roles of the observed/modeled plume height on AOD reasonably. Higher plume height will accelerate dilution near the source, as well as enhance the transportation to the downwind regions.

Fig. 3 shows that the modeled plume heights by B69 and S12 are quite close to each other for the four studied cases (different wind scenarios). In Fig. 6, the positive values of S12-B69 near $120^{o}$ W above 6 km (Fig. 6n) seems to be mainly contributed by organic carbon (Fig. 6b). Given the primary fire emissions among three plume rise models are kept being the same and the plume height modeled by B69 and S12 are quite similar, this "positive-negative-positive" pattern in a bottom-up or top-down direction near $120^{o}$ W can be also caused by variated sensitivity responses or modeling performances between B69 and S12 models to the fire cases in the zonal averaging region but not included in the fire case studies in the previous sections.

*Line 336:* "When comparing the CMAQ AOD to VIIRS AOD (Figures 7b-d), we applied VIIRS AOD saturation level (5) to CMAQ AOD results."

What does "saturation level (5)" here stand for?

**Other revision suggestions:**

Author affiliation information 1: space between "VA" and "22030".

Figure 10: Caption is in blue color.

---

## Author Response (AR2)

*We thank the two reviewers for their insightful comments on our manuscript. Our responses to each of the reviewer's comments are provided below in italics.*

Reviewer 1:
The authors worked hard on the new version of the manuscript considering comments from all reviewers. I believe the manuscript is in much better shape now. Most of my remaining concerns are related to smoke height evaluation, more specifically about the model variable that's used to evaluate against observations and regarding addition of analysis and discussion of the results. More details below.

Comments by line (line numbers based om tracked changes doc):

288. The sentence "The simulated plume heights were interpolated to the MISR observation MISR pixels using the nearest neighbor approach." is not clear. Does the closest prediction of injection height is mapped to the MISR/CALIPSO data? Or are smoke heights calculated from the model 3D fields of aerosols within the model used to calculate smoke heights for the model gridcells that are closest to the retrievals (for instance, like in Ye et al., 2021)? Since you are including large fires that could span multiple model grid cells it is recommended that smoke height is derived from the model 3D fields as the result will be a combination of the injection from different grid cells, and not necessarily from the closest one. Also, does this approach takes into consideration the 1.5 factor applied to the heights inside CMAQ? This would be naturally taken into consideration when using the 3D smoke fields to derive height. Also, even though fresh plumes are analyzed here, they are considered up to 100 km away from the fire, so these injections would correspond to those happening on earlier hours. Again, using 3D fields would naturally take care of this.

*Thanks for the comment. It is the smoke height from the model 3D fields. The original line 288 has been changed to:*

*"The smoke heights from model 3-dimensional fields were interpolated to the MISR observation pixels using the nearest neighbour approach."*

289. The fire size formulation in Archer-Nicholls et al., (2015) is a function of land use cover. Is this the case for the formulation included here? If so, that would make the Freitas scheme the only one to use land use info in this work (based on the info provided in the paper). If so this could be a reason why this scheme tends to behave a bit different than the other two in this work in the study cases of section 3.1. So it would be informative to add land use cover % for the fires contributing to the injection height to assess, it might explain some of the differences.

*Thanks for the comments. Equation 6 in Archer-Nicholls et al. (2015) calculates the fire size for different land use types in one grid cell. The fire size we used from RAP-Chem is the total fire size for all land use types in one grid cell. So, the land use type does not affect the results.*

302-304. I think the authors could do a much better job at discussing these results. Reference pictures of the fires and PBL height were added to Figure 3 (which are greatly appreciated) but they are not included in the discussion. MISR often shows a large spread likely due to a

combination of smoke that doesn't escape the boundary layer and free-tropospheric injections. Note that WRF has been found to often underpredict PBL height so when MISR heights get close then is likely that those are within the PBLH in reality. If injection heights predictions are used in this comparison (and not smoke heights, see my comments above) then it would be hard for the model to capture the boundary layer smoke height that can naturally occur at the same time as free-tropospheric injections in a given scene given lower winds and larger lateral dispersion occurring in the boundary layer. The visible images are instructive to show different features, like in a) the plume that's further away and is less thick is likely within the PBL, or in b,f) the deep plume within 20-30 km away from the fire seems to be associated to the easterly plume rather than the westerly plume, and in d) the most of the low values close by the fire are due to background smoke in the PBL as the elevated plume is very thin as seen in the visible image in c). I think another important point to take from Figure 3 is that modeled injections are always into the free-troposphere, which doesn't seem to be always the case for MISR. The article below was recently published and talks about this issue

Thapa, L. H., Ye, X., Hair, J. W., Fenn, M. A., Shingler, T., Kondragunta, S., Ichoku, C., Dominguez, R., Ellison, L., Soja, A. J., Gargulinski, E., Ahmadov, R., James, E., Grell, G. A., Freitas, S. R., Pereira, G., and Saide, P. E.: Heat flux assumptions contribute to overestimation of wildfire smoke injection into the free troposphere, Communications Earth & Environment, 3, 236, 10.1038/s43247-022-00563-x, 2022.

*Thanks for the comments. The plume height shown in Figure 3 is not the calculated plume injection height but is smoke height. The following discussion is added to line 264. The above paper was cited and added to the reference list.*

*"The simulated PBL heights were displayed in Figure 3 as a reference. When the fire injection height is lower than the PBL height, the pollution could become confined in the PBL (Sofiev et al. 2021; Thapa et al, 2002). However, when the plume height is higher than PBL, the fire smoke can be dispersed into the free troposphere where wind speeds are stronger, leading to a wider range of pollution dispersion. In all four cases analyzed in Figure 3, the simulated plume heights from the three schemes surpassed the model PBL."*

340-343. Please be more quantitative in the evaluation. Measures of bias, error and correlation might be reasonable to use here.

*Thanks for the comment. The mean bias of each scheme was added to the paper:*

*"The mean bias for the three schemes is -0.60 for B69, -0.67 for F07, and 0.13 for S12."*

Figure 4. Similarly to the MISR analysis, it would be great to have an idea of the dates, times, fires and distance from the fires (and any other useful information like the fuel type, see previous comments) that were used in this analysis. I think a table could be included as an additional panel to this figure with this information. This could help with the interpretation of these results that currently mostly describe what's happening but it's not clear why these results are occurring. For instance, these results are very different compared to those for the Milepost 21 where the Freitas scheme had deeper injections.

*Thanks for the comment. The main focus of this study is to evaluate the impact of plume injection heights on the near source and downwind air quality. More details about the comparison of the three plume rise schemes go beyond the scope of this paper. The main objective of our study is to evaluate the impact of plume injection heights on near-source and downwind air quality, and not to compare or evaluate different plume rise schemes. The CALIPSO plume injection height is from the automatic detection method not from any detailed case study. Therefore, the detailed information for each case is quite unknow to us at this stage. We agree that a more comprehensive comparison of the simulated plume height would be an interesting avenue for future research, but it falls beyond the scope of our current study. Thank you again for your valuable feedback.*

Section 3.1-3.2. I think sections 3.1 and 3.2 are a bit disconnected, which is understandable as it goes from case studies to monthly means, but I think a simple plot could be added to better connect them. The authors could include box and whisker plots of the Aug-Sept 2020 mean of modeled injection heights for the different schemes. That could allow to see how representative the behavior of the case studies are compared to the means, and will also allow to verify the behavior described on section 3.2 regarding how B69 tends to always have deep injections but the other two schemes have more variability with a lower average but often very high injections for extreme cases.

*Thanks for the comments. New figure 5 and the following discussion have been added to section 3.2:*

*"Figure 5 shows the vertical profile of the two-month average $PM_{2.5}$ concentration for the three experiments. Over the two months, B69 simulated a higher average plume height and injected more $PM_{2.5}$ in the free troposphere than F07 and S12. Meanwhile, F07 simulated a lower average plume height and therefore keep more $PM_{2.5}$ in the boundary layer than B69 and S12.*

[Figure]

*Figure 5: Vertical profile of two-month average PM2.5 concentration for B69, F07, and S12 in the CONUS domain."*

419-422. AOD underprediction by models is commonly found for large wildfire events which is consistent with the result on this study, so might want to cite some of these articles here.

*Thanks for the comments. The AOD prediction performances for different models are different. Everything depends on emission factors, whether FRP or burnt area are used as proxies, and whether the satellite manages to see the dense parts of the plume without misinterpreting them as clouds. Sometimes AOD is underestimated. For instance, this work:*
*Toll, V., Reis, K., Ots, R., Kaasik, M., Männik, A., Prank, M., and Sofiev, M.: SILAM and MACC reanalysis aerosol data used for simulating the aerosol direct radiative effect with the NWP model HARMONIE for summer 2010 wildfire case in Russia, Atmospheric Environment, 121, 75–85, https://doi.org/10.1016/j.atmosenv.2015.06.007, 2015.*
*So, we decide not to cite papers here.*

Figure 8. A reference map of the photolysis rate reduction by smoke for one of the schemes (B69 might be convenient as is being used as reference) would be helpful to interpret results. Adding 2 panels to this figure (for absolute and relative differences) might work the best. The differences in photolysis rates due to smoke can then be compared to the other studies cited to check if they are within the same ranges or not. Since the current plots are differences within scheme then this comparison to other studies is not possible.

*Thanks for the comment. The maps of the absolute and relative photolysis rate reduction by smoke for B69 have been added to new figure 9. The following discussions have been added to line 405:*

*"Figures 9 a and d show the photolysis rate difference and difference ratio between B69 and the NoFire experiments. The photolysis rate results in the B69 were lower than the NoFire simulation, which proves that fire smoke led to the reduction of the photolysis rate, consistent with the findings of previous studies. The photolysis rate reduction caused by the fire smoke was found in the whole domain, both in the near-source region and the downwind region. Near the fire source, the photolysis rate reduction was more than 50%. Figures b, c, e, and f show the photolysis rate difference and difference ratio between the three experiments with different plume rise schemes. Near the source region where F07 and S12 simulate a higher AOD than B69 (Figure 8), the NO2_IUPAC10 is reduced. Meanwhile, in the downwind region, where F07 and S12 simulate a lower AOD, the photolysis rate is higher than B69. Therefore, the difference in the plume injection height would affect the fire-induced photolysis rate reduction.*

[Figure]

*Figure 9: The average photolysis rate NO2_IUPAC10 differences between B69 and NoFire (a), between F07 and B69 (b), and between S12 and B69 (c) from August 1st, 2020 to September 30th, 2020; and the average photolysis rate NO2_IUPAC10 difference ratio B69 and NoFire (d), between F07 and B69 (e), and between S12 and B69 (f) during the same period."*

632. Not sure why "humidity" is mentioned here if it was not discussed before
*We mention humidity in section 3.1. We found that the higher F07 plume rise on August 15 may be related to high humidity on that day.*

Minor Edits

28 Add CALIPSO

*Thanks for the comments. "Cloud-Aerosol Lidar and Infrared Pathfinder Satellite Observations (CALIPSO)" has been added to line 28.*

35. Since you are not running in forecasting mode I would replace "forecasts" by "predictions"
*Thanks for the comment. "forecast' has been changed to "prediction" in line 36.*

105-107. Section numbers need to be updated
*Thanks for the comment. The section numbers have been updated.*

331-338. This whole explanation belongs to the methods
*Thanks for the comment. Line 331-338 is moved to section 2.4.1.*

*"CALIPSO is an Earth Science observation mission that was launched on 28 April 2006 and flies in a nominal orbital altitude of 705 km and an inclination of 98 degrees as part of a constellation of Earth-observing satellites. CALIPSO's lidar instrument, the Cloud-Aerosol Lidar with Orthogonal Polarization (CALIOP), provides high-resolution vertical profiles of aerosol and cloud attenuated backscatter signals at 532 nm and 1064 nm (Winker et al., 2007). The footprint of the lidar beam has a 100 m cross-section with an overpass around 1:30 p.m. local time. The CALIPSO smoke injection heights are directly calculated from Level 1 attenuated backscatter profiles at 532 nm following Amiridis et al. (2010). There are several steps involved in this process. First, GBBEPx FRP data were used to locate the fire location along the CALIPSO swath. Then, a slope method (Pal et al., 1992) is applied to each profile to smooth out the original Level 1 532 nm attenuated backscatter coefficient profiles at each fire point. Next, we calculate the steep gradient in the attenuated backscatter profiles. The height of the minimum gradient value is selected as the smoke injection height."*

283. Add citation for this algorithm, I believe is the following:
Kondragunta, S.; Laszlo, I.; Ma, L. JPSS Program Office (2017): NOAA JPSS Visible Infrared Imaging Radiometer Suite (VIIRS) Aerosol Optical Depth and Aerosol Particle Size Distribution Environmental Data Record (EDR) from NDE. [NOAA-20 dataset]. NOAA National Centers for Environmental Information. NOAA Natl. Cent. Environ. Inf 2017.

*Thanks for the comment. The above paper is added to section 2.4.3 and the reference list.*

332. Can you clarify that only daytime retrievals were used?
*Thanks for the comment. Original line 332 (new 286) has been changed to:*
*"The vertical profiles of CMAQ simulated $PM_{2.5}$ are also compared to the CALIPSO daytime aerosol vertical profile."*

394. I think this should be "Fig 3e" instead of 3a.
*Thanks for the comment. "Fig 3a" has been changed to "Fig 3e" in the original line 394 (new 353).*

630. add CALIPSO

*Thanks for the comment. Line original line 630 (new 494) was changed to:*
*"The plume heights simulated by all three schemes are comparable to MISR and CALIPSO observations of aerosol height."*

668. I think the NO2 analysis was removed so I would remove this sentence.
*Thanks for the comment. The original line 668 has been removed.*

Reviewer 2:
The revised manuscript has addressed the reviewers' comments and provided additional analysis to improve the demonstration. Further comments and discussion to the revised manuscript is provided below:
Minor revision suggestions:
Fig. 6
Fig 6 shows the zonal averaged concentrations throughout the two-month period to illustrate how different plume rise models affect the spatial and vertical distribution of the studied atmospheric species, and species' fractional contribution in PM2.5.
The occurrence of wildfires in the US is quite sporadic. The zonal averaging is appropriate to highlight the dominant pattern of the fire activities in the US. What is the latitude range you've applied for the zonal averaging? Are you using a constant averaging range (from the west to the east) or a changing averaging range as the smoke dispersed?

*Thanks for the comments. The zonal average is for the whole CONUS domain, we use the same zonal average region for all the analyses. To make it clear, "over the whole domain" has been added to line 342.*

Result analysis of Fig. 6 (Section 3.3)
Authors explained the roles of the observed/modeled plume height on AOD reasonably. Higher plume height will accelerate dilution near the source, as well as enhance the transportation to the downwind regions.
Fig. 3 shows that the modeled plume heights by B69 and S12 are quite close to each other for the four studied cases (different wind scenarios). In Fig. 6, the positive values of S12-B69 near 120o W above 6 km (Fig. 6n) seems to be mainly contributed by organic carbon (Fig. 6b). Given the primary fire emissions among three plume rise models are kept being the same and the plume height modeled by B69 and S12 are quite similar, this "positive-negative-positive" pattern in a bottom-up or top-down direction near 120o W can be also caused by variated sensitivity responses or modeling performances between B69 and S12 models to the fire cases in the zonal averaging region but not included in the fire case studies in the previous sections.

*Thanks for the comments. To show the overall plume height simulation of the three schemes, a new figure, Figure 5, and the following discussions have been added to the paper.*

*". Figure 5 shows the vertical profile of the two-month average $PM_{2.5}$ concentration for the three experiments. Over the two months, B69 simulated a higher average plume height and injected more $PM_{2.5}$ in the free troposphere than F07 and S12. Meanwhile, F07 simulated a lower average plume height and therefore keep more $PM_{2.5}$ in the boundary layer than B69 and S12.*

[Figure]

*Figure 5: Vertical profile of two-month average PM2.5 concentration for B69, F07, and S12 in the CONUS domain."*

Line 336: "When comparing the CMAQ AOD to VIIRS AOD (Figures 7b-d), we applied VIIRS AOD saturation level (5) to CMAQ AOD results."
What does "saturation level (5)" here stand for?

*Thanks for the comments. For VIIRS data, it has a saturation level for AOD (e.g., AOD=5 for VIIRS). All points with an AOD higher than this saturation level was changed to the saturation level. To make this clear, original line 336 (new 374) was changed to:*

*"...we applied VIIRS AOD saturation level (AOD ≤5) to CMAQ AOD results (any CMAQ AOD values higher than 5 were changed to 5)"*

Other revision suggestions:
Author affiliation information 1: space between "VA" and "22030".
*Thanks for the comment. Space has been added between "VA" and "22030".*

Figure 10: Caption is in blue color.
*Thanks for the comment. The caption for figure 10 has been changed to black.*